# Reward Poisoning in Reinforcement Learning: Attacks Against Unknown Learners in Unknown Environments

## Abstract

We study black-box reward poisoning attacks against reinforcement learning (RL), in which an adversary aims to manipulate the rewards to mislead a sequence of RL agents with *unknown* algorithms to learn a nefarious policy in an environment *unknown* to the adversary a priori. That is, our attack makes minimum assumptions on the prior knowledge of the adversary: it has no initial knowledge of the rewards, transitions, or the learner, and neither does it observe the learner's internal mechanism except for its performed actions. We design a novel black-box attack, U2, that can provably achieve a near-matching performance to the state-of-the-art white-box attack, demonstrating the feasibility of reward poisoning even in the most challenging black-box setting.

## 1 Introduction

*Reward poisoning* refers to an adversarial attack against reinforcement learning (RL) where the adversary manipulates the rewards in order to mislead the RL agent's learning process. It has been considered by many as a realistic threat against modern RL applications. Many real-world applications—such as recommendation systems (Zhao et al., 2018; Chen et al., 2019), virtual/conversational assistants (Dhingra et al., 2016; Li et al., 2016)—extract reward signals directly from user feedback and are thus prone to adversarial corruption. Reward poisoning has recently been study in various settings (Zhang & Parkes, 2008; Zhang et al., 2009; Ma et al., 2018; Jun et al., 2018; Peltola et al., 2019; Altschuler et al., 2019; Liu & Shroff, 2019; Ma et al., 2019; Huang & Zhu, 2019; Rakhsha et al., 2020a;b; Zhang et al., 2020c), but most of the prior work makes strong assumptions on the knowledge of the adversary. It is often assumed that the adversary has full knowledge of the environment (i.e., true rewards/transitions) or the agent's learning algorithm or both. Under such assumptions, attack strategies have been proposed that can mislead the agent to learn a nefarious policy with minimal perturbations to the rewards.

However, in many applications, the adversary has limited knowledge about the environment or the agent's learning algorithm. For example, on e-commerce platforms, the adversary may take the form of a malicious seller who wants to mislead the platform's ranking system to promote their product by posting fake purchases, reviews, or comments. In such scenarios, the malicious sellers often have very limited knowledge about the dynamics of the market or the particular ranking algorithm currently used by the platform. In this setting, the attack strategies developed by prior works cannot be applied, and therefore one can argue that the security threats anticipated in these works might be pessimistic.

To evaluate the security threat against RL agents in more realistic scenarios, in this work, we investigate the *unknown-unknown* attack setting, where we assume that the adversary has *no knowledge* about the environment or the agents' learning algorithm. To the best of our knowledge, we are the first to study theoretically sound reward-poisoning attacks against RL in this setting. Our contributions are three folds.

1. We develop a black-box attack strategy, U2, that can attack unknown RL agents (learners) in an unknown environment. U2 operates without any prior knowledge of the rewards, transitions, or the learners and only requires that learners follow a no-regret RL algorithm.
2. We show that surprisingly, with appropriate choice of hyperparameters, U2 can achieve an attack cost not much worse than the optimal white-box attack (Rakhsha et al., 2020a;b).

3. As part of the U2 attack, we develop an exploration subroutine that is of independent interest. This subroutine can turn any no-regret RL algorithm to apply to the reward-free/task-agnostic (Jin et al., 2020b; Zhang et al., 2020b) RL settings, where the learner is manipulated to explore the whole state-action space and allow data to be efficiently collected for arbitrary down-stream tasks (in our case the poisoning attack task).

**Related Work.** Reward poisoning against RL has been first studied in *batch RL* (Zhang & Parkes, 2008; Zhang et al., 2009; Ma et al., 2019) where rewards are stored in a pre-collected data set by some behavior policy, and the attacker comes in to modify the batch data. Because all data are available to the attacker at once, the batch attack problem is somewhat easier. Our paper instead focuses on the *online* RL attack setting where reward poisoning must be done on the fly. In online settings, reward poisoning is first introduced and studied in multi-armed bandits (Ma et al., 2018; Jun et al., 2018; Peltola et al., 2019; Altschuler et al., 2019; Liu & Shroff, 2019), where the authors show that adversarially perturbed reward can mislead standard bandit algorithms to pull a suboptimal arm or suffer large regret.

Online reward poisoning attacks in the white-box setting have been studied (Huang & Zhu, 2019; Rakhsha et al., 2020a;b; Zhang et al., 2020c), where the adversary is assumed to have full knowledge of the MDP or the learning algorithm. Among them, (Huang & Zhu, 2019; Rakhsha et al., 2020a;b) focus on attacking the reward function itself, in which case the adversarial rewards are also functions of state and action, but independent of the learning process. (Zhang et al., 2020c) focuses on attacking a Q-learning agent, and presents a more powerful attack that can depend on the RL victim's Q-table $Q_t$. Their analysis shows that such adaptive attacks can be exponentially faster in enforcing the target policy than non-adaptive attacks studied in prior works. In comparison, our work focus on the more challenging black-box setting, and our attack can be applied to any no-regret RL algorithms. Recently, (Xu & Rabinovich) has studied black-box attacks in which the attacker modifies the environment parameters, which is a different setting and lacks any theoretical analysis compared to this paper. In another work, (Sun & Huang, 2020) empirically studied the problem of black-box poisoning attack against policy-based deep RL algorithms. Their algorithm VA2C-P takes an actor-critic structure and shows strong attack performance against state-of-the-art policy gradient algorithms, such as REINFORCE, A2C, PPO, etc. In comparison, our work provides a more general and theoretically sound black-box attack strategy against any efficient RL algorithms, including policy gradient algorithms.

## 2 Problem Setup

### 2.1 Preliminaries and Definitions

In this work, we assume that the environment is modeled as a Markov Decision Process (MDP), defined by a tuple $M = (S, A, R, P, s^0, \gamma)$, where $S$ is a finite state space, $A$ is a finite action space, $R\colon S \times A \to \mathbb{R}$ is a reward function, $P\colon S \times A \times S \to [0, 1]$ is the transition probability function, $s^0$ is the initial state, and $\gamma \in (0, 1)$ is the discounting factor. To simplify the notation, from here on, we will omit $s^0, \gamma$ and denote $M = (S, A, R, P)$; when clear from context, we will abuse the notation and also use $S, A$ to denote the size of the state space and action space respectively. At the beginning, the agent starts from the initial state $s^0$. By taking action $a$ at state $s$, the agent receives a reward with expectation $R(s, a)$ and $\sigma^2$-sub-Gaussian noise for some $\sigma > 1$, and transits to state $s'$ with probability $P(s, a, s')$.

A (deterministic) *policy* is a mapping from states to actions, i.e., $\pi\colon S \to A$. We will use the standard state value function $V_M^\pi(s) = \mathbb{E}\left[\sum_{\tau=0}^\infty \gamma^\tau r_\tau | s_0 = s, \pi\right]$ and the state-action value function $Q_M^\pi(s, a) = \mathbb{E}\left[\sum_{\tau=0}^\infty \gamma^\tau r_\tau | s_0 = s, a_0 = a, \pi\right]$ where the expectations are over the stochasticity in both transition and reward functions. The optimal value functions are also defined as $Q_M^*(s, a) = \max_\pi Q_M^\pi(s, a)$ and $V_M^*(s) = \max_\pi V_M^\pi(s)$. The expected discounted reward $\rho_M^\pi$ of policy $\pi$ is defined as $\mathbb{E}\left[\sum_{\tau=0}^\infty \gamma^\tau r_\tau | \pi\right]$ which gives $\rho_M^\pi = V_M^\pi(s^0)$. Policy $\pi^*$ is said to be optimal if $\rho_M^{\pi^*} \geq \rho_M^\pi$ for every $\pi \neq \pi^*$ and $\varepsilon$-robust optimal if $\rho_M^{\pi^*} \geq \rho_M^\pi + \varepsilon$ also holds. We denote the expected discounted reward of the optimal policy by $\rho_M^*$. A policy $\pi$ is called $\varepsilon$-optimal if it is at most $\varepsilon$ worse than the optimal policy, i.e. $\rho_M^\pi \geq \rho_M^* - \varepsilon$, and $\varepsilon$-suboptimal otherwise. A step is called $\varepsilon$-suboptimal if the action performed is $\varepsilon$-suboptimal, i.e., $a_\tau \notin \{\pi(s_\tau) : \rho_M^\pi \geq \rho_M^* - \varepsilon\}$ (action is not chosen by any $\varepsilon$-optimal policy). Let $\mu^\pi$ be the (unnormalized)

state distribution of policy $\pi$ defined as $\mu_M^\pi(s) = \sum_{\tau=0}^\infty \gamma^\tau \mathbb{P}[s_\tau = s|\pi]$. Let $\mu_{\min} = \min_{s,\pi} \mu_M^\pi(s)$. We assume under any policy $\pi$, all states are visited with a positive probability, i.e. $\mu_{\min} > 0$. Finally, for policy $\pi$ and a state-action pair $(s, a)$ we define the neighboring policy of $\pi$ at $(s, a)$ denoted by $\pi\{s; a\}$ as

$$\pi\{s; a\}(x) = \begin{cases} \pi(x) & x \neq s \\ a & x = s \end{cases}.$$

## 2.2  Attack Problem

**Learners.**  To provide theoretical insight into teaching effectiveness. We assume that the learners are *sample-efficient*. In particular, we make the following assumption on the learner.

**Assumption 2.1** (PAC learners)**.** With probability of at least $1 - \delta$, the learner performs $\varepsilon$-suboptimal actions at most $\text{SubOpt}(T, \varepsilon, \delta)$ times where $\text{SubOpt}$ is sublinear in $T$. Moreover, for some $\alpha, \beta > 0$, the learner is able to find an $\alpha$-optimal policy in $T$ steps with probability of at least $1 - \beta$.

Assumption 2.1 is satisfied by most sample-efficient RL algorithms in the literature, such as UCRL2 (Auer et al., 2009), UCBVI (Azar et al., 2017; Dann et al., 2017), UCB-H (Jin et al., 2018), etc. In particular, one can show that a sub-linear regret is sufficient for the algorithm to satisfy both properties in Assumption 2.1.

Given this assumption, in this work, we focus on a *population learning* scenario, in which a sequence of $L$ online RL agents take turns to interact with the environment. Such scenarios are relevant when many learners aim to learn the same (or similar) task, for example, RL agents as auto-pilots for autonomous transportation systems or RL agents as virtual personal assistants that learn independently to adapt to the preferences of their users. In the era of rapid growth of edge devices, we believe that such population/federate learning scenario will become more and more common in practice, comparing to the traditional centralized learning systems. Specifically, we consider a setting where each learner interacts with the environment for a total of $T$ times. At interaction $t$ of learner $l$, the learner chooses action $a_t^{(l)}$ from state $s_t^{(l)}$, and the environment produces reward $r_t^{(l)}$ and next state $s_{t+1}^{(l)}$. The learner is moved to $s_{t+1}^{(l)}$, but the attacker changes the observed reward from $r_t^{(l)}$ to $r_t'^{(l)}$.

We note that while we focus on the population learning setting, there is no technical barrier to apply our algorithm in the single learner setting. If one is willing to make stronger assumptions on the learner, e.g. if the learner can "recover" from the change of reward function, then the exact same attack strategy can be applied to attack a single learner during the learning process. Many learners in practice exhibit such a property, and the theoretical study of such learners was done under the name of *adversarial RL*, e.g. (Rosenberg & Mansour, 2019; Jin et al., 2020a).

**Attacker.**  In this paper, we study the *black-box reward poisoning attack* problem. In this setting, the attacker has no prior knowledge about the rewards, the transitions, or the learners. The attacker only knows that the learners are *efficient*, i.e. the learners satisfy the $(\alpha, \beta)$ and $\text{SubOpt}(T, \varepsilon, \delta)$ guarantees, but does not know the actual parameters $\alpha, \beta$ and the $\text{SubOpt}$ function. The attacker has a *target policy* $\pi_\dagger$ and wants to force the learners to follow this policy by making small changes in the observed rewards. This objective is formulated by an *attack cost* function $\text{Cost}(T, L)$ defined as

$$\text{Cost}(T, L) := \frac{1}{L \cdot T} \sum_{l=1}^L \sum_{t=1}^T \left( |r_t^{(l)} - r_t'^{(l)}| + \lambda \mathbb{1}\left[ a_t^{(l)} \neq \pi_\dagger(s_t^{(l)}) \right] \right)$$

where $\mathbb{1}[.]$ denotes the indicator function. Intuitively, the attacker needs to pay the cost $|r_t^{(l)} - r_t'^{(l)}|$ to change the reward from $r_t^{(l)}$ to $r_t'^{(l)}$ and will be penalised with an additional cost of $\lambda$ for each step the learner does not follow the target policy, which we will call a *mismatch*. The final objective is defined as the average cost over all $L \cdot T$ steps. Here, $\lambda > 0$ is a parameter balancing the trade-off between the two goals. Throughout the attack process, the attacker only observes the interaction between the learners and the environment, i.e. $s_t^{(l)}, a_t^{(l)}, r_t^{(l)}$ and does not observe the internal process of learners' algorithms or the environment.

# 3 Overview of U2 and Main Results

In this paper, we introduce an attack strategy for the attacker to enforce the target policy without any knowledge about either the environment or the learners. We present an *explore-and-exploit* attack strategy, U2, and demonstrate its optimality by comparing its attack cost with the optimal white-box attack in the literature. In what follows, we first introduce a state-of-the-art white-box attack. Our black-box attack U2 builds upon this white-box attack and is introduced in the second half of the section.

## 3.1 White-box Attack

To begin with, consider the white-box attack problem, in which the attacker has full knowledge of the MDP and the learner. White-box attacks have been studied extensively in the literature (Huang & Zhu, 2019; Rakhsha et al., 2020a; Zhang et al., 2020c). Here, we will utilize a state-of-the-art attack method that is agnostic to the learning algorithm and hence is suitable for designing U2. Below, we briefly summarize the intuition behind this method. The key idea behind the attack is to design the poisoned rewards $r_t^{\prime(l)}$ to come from a reward function $R'$ such that $\pi_\dagger$ is $\varepsilon$-robust optimal in $M' = (S, A, R', P)$. This way, the only $\varepsilon$-optimal policy will be the target policy, and all the steps in which $\pi_\dagger$ is not followed will be $\varepsilon$-suboptimal. Thus, the learner does not follow the target policy in only sublinear number of steps in $T$.

While such a reward function can successfully force the target policy, it may incur a large attack cost because $R'$ may be different from $R$ on the target actions (actions used in the target policy). One way to avoid this is to add another constraint when designing the adversary reward $R'$ such that the rewards on the target actions remain unchanged. Consequently, the attacker will only pay the cost $\lambda + |r_t^{(l)} - r_t^{\prime(l)}|$ on mismatches (the steps the target policy is not followed) which will only happen a number of times sublinear in $T$ if the agent is a no-regret RL learner. Specifically, this adds a constraint $R'(s, \pi_\dagger(s)) = R(s, \pi_\dagger(s))$ for every state $s$. Letting $R' = R - \Delta$ for some $\Delta: S \times A \to \mathbb{R}$, this attack can be performed by setting

$$r_t^{\prime(l)} = r_t^{(l)} - \Delta(s_t^{(l)}, a_t^{(l)}) \quad \forall l, t. \tag{1}$$

One can bound the cost of this attack using the $\text{SUBOPT}(T, \varepsilon, \delta)$, obtaining the following result:

**Lemma 3.1.** *Assume the attacker performs the attack described in* (1) *for some* $\Delta: S \times A \to \mathbb{R}$ *on all the learners. Then, with probability of at least* $1 - \delta$,

$$\text{COST}(T, L) \leq (\|\Delta\|_\infty + \lambda) \cdot \frac{\text{SUBOPT}(T, \varepsilon, \frac{\delta}{L})}{T}$$

The bound follows directly from the fact that the attacker incurs no cost on steps when $a_t^{(l)} = \pi_\dagger(s_t^{(l)})$ and the cost on other steps which are at most $L \cdot \text{SUBOPT}(T, \varepsilon, \frac{\delta}{L})$ steps, is at most $\|\Delta\|_\infty + \lambda$. The problem of finding the $\Delta$ that minimizes the upper bound in Lemma 3.1 can be formulated as the following program:

$$
\begin{aligned}
\min_{\Delta} \quad & \|\Delta\|_\infty & \text{(P1)} \\
\text{s.t.} \quad & \pi_\dagger \text{ is } \varepsilon\text{-robust opimal in } (S, A, R - \Delta, P) \\
& \forall s: \quad \Delta(s, \pi_\dagger(s)) = 0.
\end{aligned}
$$

If the attacker has full knowledge of the MDP, it can directly solve for (P1), which has been shown to have a closed-form solution (Rakhsha et al., 2020a):

$$\Delta_M^*(s, a) = \left[ Q_M^{\pi_\dagger}(s, a) - V_M^{\pi_\dagger}(s) + \frac{\varepsilon}{\mu_M^{\pi_\dagger\{s;a\}}(s)} \right]^+$$

for $a \neq \pi_\dagger(s)$, and $\Delta_M^*(s, \pi_\dagger(s)) = 0$ for every $s$. Here, $[x]^+ = \max(0, x)$. For completeness, we state the above result in the following lemma:

---

**Algorithm 1** U2

---

**Input:** $S, A, \gamma, \sigma, p, \varepsilon, m$
**Initialize:** $l \leftarrow 1$
**Exploration Phase:**
  **repeat**
    **for** $t = 1$ **to** $T$ **do**
      Set $r_t'^{(l)} \sim \text{Uniform}\{-1, 1\}$.
    Calculate confidence set $\mathcal{M}$ of possible MDPs.
    $l \leftarrow l + 1$
  **until** $\mathcal{M}$ satisfies condition (7).
 **Attack Phase:**
  Solve (P2) to get $\widehat{\Delta} \colon S \times A \to \mathbb{R}$
  **while** $l \leq L$ **do**
    **for** $t = 1$ **to** $T$ **do**
      Set $r_t'^{(l)} = r_t^{(l)} - \widehat{\Delta}(s_t^{(l)}, a_t^{(l)})$.
    $l \leftarrow l + 1$

---

**Lemma 3.2.** *(Rakhsha et al., 2020a). The optimal solution for* (P1) *is* $\Delta_M^*$*. Moreover,* $\Delta$ *is a feasible solution of* (P1) *if and only if for every state-action pair* $s, a$*,* $\Delta(s, a) \geq \Delta_M^*(s, a)$ *and* $\Delta(s, \pi_\dagger(s)) = 0$*.*

Note that Lemma 3.2 implies that the choice of norm in the objective of (P1) can be arbitrary, e.g. $L_1$, $L_2$ or $L_\infty$ norms, while the optimal solution $\Delta_M^*(s, a)$ remains unchanged, showing the robustness of this solution. This optimal white-box attack serves as a baseline for our black-box attack strategy.

### 3.2 Black-box Attack

In this work, however, we study the problem in which the attacker has no knowledge of the MDP's rewards/transitions, and thus can no longer directly solve (P1) to perform the attack. In this setting, we propose an attack strategy that consists of two separate phases.

**Exploration phase.** To begin with, the attacker aims to collect data on the environment by providing rewards that encourage the learners to explore the whole MDP. This goal is achieved by providing the following simple yet effective rewards:

$$r_t' \sim \text{Uniform}\{-1, 1\} \tag{2}$$

We will show that this simple reward function enforces the learner to provably visit all $(s, a)$ pairs sufficiently often and allows the attacker to learn about the MDP rewards and transitions to perform the attack. We discuss the guarantee and the intuition of this simple reward function in Section 4.1.

**Attack phase.** Once the attacker has gathered enough observations, it can start to attack the rest of the learners by estimating a set of plausible MDPs $\mathcal{M}$. It then solves for a robust perturbation $\Delta$ that is guaranteed to enforce the target policy $\pi_\dagger$ on all $M \in \mathcal{M}$. This robust attack problem can be formulated as a robust version of problem (P1):

$$\min_\Delta \quad \|\Delta\|_\infty \tag{P2}$$
$$\text{s.t.} \quad \forall (S, A, \widetilde{R}, \widetilde{P}) \in \mathcal{M} :$$
$$\pi_\dagger \text{ is } \varepsilon\text{-robust optimal in } (S, A, \widetilde{R} - \Delta, \widetilde{P})$$
$$\forall s : \quad \Delta(s, \pi_\dagger(s)) = 0.$$

In Section 4.2, we describe the process of solving (P2) in detail. Following the two-phase procedure, the attack cost of U2 can be upper bounded by the following theorem:

**Theorem 3.1.** *For any $m > 0$ and $p \in (0, 1)$, assume that $\alpha < \frac{\mu_{min}}{2\sqrt{2}}$ and $\beta < \frac{1}{8SA}$, then, with probability of at least $1 - 4p$, the cost of* U2 *is bounded by*

$$\frac{k_0}{L} \cdot \left( \|R\|_\infty + \sigma \sqrt{2 \log \frac{2k_0 T}{p}} + 1 + \lambda \right) + (\|\Delta_M^*\|_\infty + \lambda + m) \cdot \frac{\text{SubOpt}(T, \varepsilon, \frac{p}{L})}{T}$$

*where $k_0$ is a function of MDP $M$, $p$, $\alpha$, $\beta$, $\lambda$, $m$, $\varepsilon$, and $L$ as defined in (8).*

The given bound on $\text{Cost}(T, L)$ consists of two terms. The first and the second term are bounds for the cost of the exploration phase and the attack phase, respectively. In the first term, $k_0$ is of the order $O(1 + \frac{\alpha^2 \log L}{m^2})$. Here, $m$ is a hyperparameter of U2 that dictates how closely the attack phase cost should match with the optimal white-box attack. The value of $\alpha$ can be chosen to be smaller for larger values of $T$, and a fixed $\beta$. This dependence on $T$ is based on the sample complexity of the learner's algorithm. For an algorithm such as (Wang et al., 2020) that the sample complexity matches the lower bound in the dependence on $\alpha$, one can choose $\alpha = O(\frac{1}{\sqrt{T}})$, which leads to $k_0 \in O(1 + \frac{\log L}{m^2 \cdot T})$. Thus, the first term has an order of $\tilde{O}(\frac{1}{L} + \frac{1}{m^2 \cdot T \cdot L})$ disregarding the logarithmic terms and diminishes for large enough $L$.

The second term is $m \cdot \text{SubOpt}(T, \varepsilon, \frac{p}{L})/T$ worse than the attack cost achievable by the optimal white-box attack as in Lemma 3.2. The second term is diminishing in $T$ due to the assumption that $\text{SubOpt}(T, \varepsilon, \frac{p}{L})$ is sublinear in $T$. Therefore, U2 can be viewed as a *no-regret* attack strategy, whose averaged attack cost diminishes to zero as $T$ and $L$ go to infinity, as is achieved by the attack of (Rakhsha et al., 2020a) in the white-box setting. Note that even though the optimal value of hyperparameter $m$ to minimize the bound cannot be found without the knowledge of the learner's sample complexity bounds, this asymptotic behavior of algorithm holds for any value of $m$.

## 4 Technical Details of U2

In what follows, we present the details of the U2 strategy in both the exploration and attack phase.

### 4.1 Exploration Phase

In this phase, the goal is to collect observations on the MDP to estimate its parameters, which will be used in the attack phase to find an effective reward perturbation. With more observations, the attacker will be able to build a smaller confidence set $\mathcal{M}$ on the environment MDP and find a $\widehat{\Delta}$ of smaller norm. In the extreme case where the attacker gathers an infinite number of observations on all $(s, a)$ pairs, it can find the optimal $\Delta_M^*$ and match the optimal white-box attack.

In order to gather observations on all $(s, a)$ pairs, the attacker needs to design adversarial rewards that encourage the learners to explore the environment. Our key observation is that, despite not knowing anything about the MDP or the learner, the attacker can still utilize the learners' learning guarantee to provably collect observations. The idea is to draw $r_t'$ from a reward function $R_E$, such that finding a nearly optimal policy in $M_E = (S, A, R_E, P)$ requires properly exploring all states and actions. This condition can be met by choosing $R_E$ in a way that the gap of the optimal Q function is small, i.e. the $Q_{M_E}^*(s, a)$ values are similar for different actions $a$. Specifically, we show that the uniform Bernoulli reward function in (2) effectively enforces the learner to explore, as detailed in the following lemma.

**Lemma 4.1.** *Let $s, a$ be an arbitrary state and action pair, and $g(s, a) = \min_{\pi:\pi(s)=a} \mu_M^\pi(s)$. Assume $4\beta \le \delta$ and $\frac{\alpha}{g(s,a)} < \frac{1}{2\sqrt{2}}$. If the feedback as in (2) is given to a learner, then at the end of $T$ steps, with probability of at least $1 - \delta$, action $a$ is chosen from state $s$ for at least*

$$\frac{g(s, a)^2}{\alpha^2} \cdot \frac{c_1 \cdot (\log \frac{\delta}{4\beta})^2}{\log \frac{8}{\delta} + c_2 \cdot (\log \frac{\delta}{4\beta})}$$

*number of times, where $c_1 = 0.08$ and $c_2 = 1.34$.*

Lemma 4.1 provides a lower bound on the number of data points that will be collected by each learner under our exploratory reward function. Note that the bound is true for any values of $\alpha, \beta$ that Assumption 2.1 holds for. Thus, for a fixed $T$ one can find an $(\alpha, \beta)$ with very small $\alpha$ but large $\beta$, to obtain a larger lower bound. The detailed proof of this lemma is deferred to the Appendix.

After each learner's interaction in the exploration phase, the attacker builds a confidence set $\mathcal{M}$ for plausible environment MDPs for two main purposes. First, these sets are used in the exploration phase to check whether the gathered data is enough for the attack phase and the attacker can stop the exploration. Second, the last set is used in the attack phase to find an effective attack in the environment. After each learner, let $N(s, a)$ be the number of times state-action pair $(s, a)$ is observed, and let $N_{\min} = \min_{s,a} N(s, a)$. If $N_{\min} > 0$, we define the following empirical estimates of $R(s, a)$ and $P(s, a, s')$:

$$\widehat{R}(s, a) = \frac{\sum_{t,l} \mathbb{1}\left[s_t^{(l)} = s, a_t^{(l)} = a\right] \cdot r_t^{(l)}}{N(s, a)}$$

$$\widehat{P}(s, a, s') = \frac{\sum_{t,l} \mathbb{1}\left[s_t^{(l)} = s, a_t^{(l)} = a, s_{t+1}^{(l)} = s'\right]}{N(s, a)}$$

We will also define the following confidence sets of reward $\mathcal{R}(s, a)$ and transition $\mathcal{P}(s, a)$ as

$$\mathcal{R}(s, a) = \left\{ r \in \mathbb{R} : |r - \widehat{R}(s, a)| \leq \frac{u}{\sqrt{N(s, a)}}) \right\}$$

$$\mathcal{P}(s, a) = \left\{ d \in \Lambda(S) : \|d - \widehat{P}(s, a, \cdot)\|_1 \leq \frac{w}{\sqrt{N(s, a)}} \right\}$$

where $\Lambda(S)$ is the probability simplex over $S$ and

$$u = \sqrt{2\sigma^2 \log(2SAL/p)},$$
$$w = \sqrt{2 \log(2SAL/p) + 2S \log 2}.$$

These two confidence intervals $u, w$ are direct consequences of Hoeffding's Inequality and (Weissman et al., 2003). Now let the confidence set $\mathcal{M}$ be the set of all MDPs $(S, A, \widetilde{R}, \widetilde{P})$ such that for every $s, a$, $\widetilde{R}(s, a) \in \mathcal{R}(s, a)$ and $\widetilde{P}(s, a, .) \in \mathcal{P}(s, a)$. This gives us the following lemma.

**Lemma 4.2.** *With probability of at least $1 - p/L$, $M \in \mathcal{M}$.*

The failure probability of $p/L$ ensures that following this scheme to build $\mathcal{M}$ after each learner, with probability of at least $1 - p$, $M$ will always be in $\mathcal{M}$. The attacker continues the exploration until $\mathcal{M}$ is small enough to perform a near-optimal attack. Since this decision involves technical details of the attack phase, we turn back to it in Section 4.3.

**Remark 4.1.** It's worth mentioning that the exploration subroutine and the guarantee in Lemma 4.1 are of independent interests to pure exploration problems in reinforcement learning. A number of recent works study the problem of task-agnostic exploration (Jin et al., 2020b; Zhang et al., 2020b), where the goal is to design a learner that can explore the MDP and collect data efficiently to prepare for any downstream task. In (Zhang et al., 2020b), their algorithm UCBZERO can be viewed as a UCB-H (Jin et al., 2018) algorithm under uniform reward, and they left as an open problem whether other no-regret algorithms can be transformed into a task-agnostic exploration algorithm. Our analysis in this section provides a positive answer. We show that any no-regret RL algorithm can provably explore all $(s, a)$ pairs in the MDP given a simple uniform reward function.

## 4.2 Attack Phase

After collecting enough data on the environment's dynamics, the attacker moves on to the attack phase in which the target policy is enforced to the remaining learners. In the beginning of the attack phase, the

attacker uses the last $\mathcal{M}$ built in the exploration phase to find an appropriate $\widehat{\Delta}$ by solving problem (P2). It then provides rewards as in (1) with $\Delta = \widehat{\Delta}$ to all the remaining learners.

To solve (P2), note that one can utilize Lemma 3.2 to rewrite the first constraint of (P2) as

$$\forall s, a: \quad \Delta(s, a) \geq \max_{\widetilde{M} \in \mathcal{M}} \Delta_{\widetilde{M}}^*(s, a). \tag{3}$$

Thus, the attacker needs to upper bound $\Delta_{\widetilde{M}}^*(s, a)$ for $\widetilde{M} \in \mathcal{M}$ to find a feasible solution for (P2). For policy $\pi$, define $V_{\text{low}}^\pi(s) = \min_{\widetilde{M} \in \mathcal{M}} V_{\widetilde{M}}^\pi(s)$ and $Q_{\text{high}}^\pi(s, a) = \max_{\widetilde{M} \in \mathcal{M}} Q_{\widetilde{M}}^\pi(s, a)$. Also let $\mu_{\text{low}}^\pi(s) = \min_{\widetilde{M} \in \mathcal{M}} \mu_{\widetilde{M}}^\pi(s)$. The attacker sets

$$\widehat{\Delta}(s, a) = \left[ Q_{\text{high}}^{\pi_\dagger}(s, a) - V_{\text{low}}^{\pi_\dagger}(s) + \varepsilon / \mu_{\text{low}}^{\pi_\dagger \{s; a\}}(s) \right]^+ \tag{4}$$

for $a \neq \pi_\dagger(s)$, and $\widehat{\Delta}(s, \pi_\dagger(s)) = 0$. By the definitions, it is clear that $\widehat{\Delta}$ satisfies condition (3), and therefore, is a feasible solution for (P2).

Computationally, the attacker can calculate quantities $V_{\text{low}}^\pi$, $Q_{\text{high}}^\pi$, and $\mu_{\text{low}}^\pi$ using *robust policy evaluation* (Iyengar, 2005; Nilim & El Ghaoui), a standard robust control procedure that is also used in no-regret model-based RL algorithms such as UCRL2 (Auer et al., 2009). Robust policy evaluation calculates the worst-case value function of a policy $\pi$, i.e. $V_{\text{low}}^\pi(s)$, when the exact model of environment is not available, and the transition distributions and rewards are just known to be in certain sets of possible values (usually confidence intervals obtained from observations). Specifically, if $R(s, a) \in \mathcal{R}(s, a)$ and $P(s, a, .) \in \mathcal{P}(s, a)$ for every $s, a$, the procedure sets $v_0^\pi(s) = 0$ for every state $s$, and applies the following iterative updates:

$$v_{i+1}^\pi(s) = \min_{r \in \mathcal{R}(s, \pi(s))} r + \gamma \min_{p \in \mathcal{P}(s, \pi(s))} \mathbb{E}_{s' \sim p} v_i^\pi(s') \tag{5}$$

Then, it is shown that $V_{\text{low}}^\pi(s) = \lim_{i \to \infty} v_i^\pi(s)$. Note that with our choices of $\mathcal{R}$ and $\mathcal{P}$, each iteration of robust policy evaluation involves $S$ special linear programming problems that can be solved in total time of $\mathcal{O}(S^2)$ (Strehl & Littman, 2008). One execution of the robust policy evaluation algorithm gives all the $V_{\text{low}}^\pi(s)$ values for $s \in S$. The same algorithm can be used to obtain values $V_{\text{high}}^\pi(s) = \max_{\widetilde{M} \in \mathcal{M}} V_{\widetilde{M}}^\pi(s)$ by substituting min with max. Then, one can set

$$Q_{\text{high}}^\pi(s, a) = \max_{r \in \mathcal{R}(s, a)} r + \gamma \max_{p \in \mathcal{P}(s, a)} \mathbb{E}_{s' \sim p} V_{\text{high}}^\pi(s')$$

Note that this computation is equivalent to (5), a single iteration of the robust policy evaluation, and therefore negligible in terms of computational cost.

Next, we show how to compute $\mu_{\text{low}}^\pi(s)$. For state $s$, define the reward function $R_s(x, a) := \mathbb{1}[x = s]$, and for $\widetilde{M} = (S, A, \widetilde{R}, \widetilde{P})$ let $\widetilde{M}_s = (S, A, R_s, \widetilde{P})$. We have $\mu_{\widetilde{M}}^\pi(s) = \rho_{\widetilde{M}_s}^\pi = V_{\widetilde{M}_s}^\pi(s^0)$. Thus, we have

$$\mu_{\text{low}}^\pi(s) = \min_{\widetilde{M} \in \mathcal{M}} \mu_{\widetilde{M}}^\pi(s) = \min_{\widetilde{M} \in \mathcal{M}} V_{\widetilde{M}_s}^\pi(s^0).$$

The value of $\min_{\widetilde{M} \in \mathcal{M}} V_{\widetilde{M}_s}^\pi(s^0)$ can again be calculated using the robust policy evaluation used for $V_{\text{low}}^\pi$, with only difference that now the reward function is known to be in the singleton set $\{R_s\}$. As one execution of the robust policy evaluation is needed for each $\mu_{\text{low}}^{\pi_\dagger \{s; a\}}(s)$ in $\widehat{\Delta}(s, a)$, the whole attack phase has time complexity of $\mathcal{O}(SA)$ runs of robust policy evaluation.

**Remark 4.2.** The robust attack procedure can also be used in the case where the attacker has a set of prior observations of the environment, and wishes to enforce the target policy to all the learners and not use any of the learners to collect more data. Such scenarios are applicable, for example, when the attacker is able to observe the natural behavior of many other learners before starting the attack. One can do similar analysis for this attack, as included in the supplementary material.

### 4.3 Conditions for Stopping Exploration

A key part of the U2 attack strategy is to decide when to end the exploration phase and start the attack phase. If the exploration is stopped earlier, the attacker is forced to make a large perturbation to the reward to compensate for the uncertainty, and thus incur a larger attack cost in the attack phase. On the other hand, a longer exploration phase allows the attacker to estimate $\Delta_M^*$ more accurately, but incurs a larger attack cost in the exploration phase.

In U2, the attacker explores the MDP until it is guaranteed that the per step cost of attack phase is at most $m$-larger then the per step cost of the optimal white-box attack, i.e. for every $s, a$

$$\widehat{\Delta}(s,a) + \lambda \le \Delta_M^*(s,a) + \lambda + m \tag{6}$$

where $m > 0$ is a hyperparameter to adjust the amount of the exploration. After each learner in the exploration phase, the attacker calculates the confidence intervals $\mathcal{R}$, $\mathcal{P}$ and then the $\mu_{\text{low}}^{\pi_\dagger\{s;a\}}(s)$ values for every $s, a$. Let $R_{\text{high}}(s,a) = \max \mathcal{R}(s,a)$ and $R_{\text{low}}(s,a) = \min \mathcal{R}(s,a)$. Define $\widehat{R}_{\text{range}} := \max_{s,a} R_{\text{high}}(s,a) - \min_{s,a} R_{\text{low}}(s,a)$ and let

$$\widehat{e}_Q := \frac{2u + 2\gamma \cdot \widehat{R}_{\text{range}} \cdot w}{(1-\gamma)^2 \cdot \sqrt{N_{\min}}} \ , \ e_\mu := \frac{2\gamma \cdot w}{(1-\gamma)^2 \cdot \sqrt{N_{\min}}}.$$

The exploration phase ends if for every state and action pair $s, a$ we have

$$2\widehat{e}_Q + \frac{\varepsilon}{\mu_{\text{low}}^{\pi_\dagger\{s;a\}}(s)} - \frac{\varepsilon}{\mu_{\text{low}}^{\pi_\dagger\{s;a\}}(s) + e_\mu} \le m \tag{7}$$

We show that given the event $M \in \mathcal{M}$, we have $V_M^{\pi_\dagger}(s) \le V_{\text{low}}^{\pi_\dagger}(s) + \widehat{e}_Q$, $Q_M^{\pi_\dagger}(s,a) \ge Q_{\text{high}}^{\pi_\dagger}(s,a) - \widehat{e}_Q$, and $\mu_M^{\pi_\dagger\{s;a\}}(s) \le \mu_{\text{low}}^{\pi_\dagger\{s;a\}}(s) + e_\mu$. Consequently, it is shown that the goal (6) is satisfied once (7) is satisfied for every $s, a$.

This stopping condition gives a simple bound on the cost of the attack phase. However, the number of learners that will be used for exploration, is also important as it decides the cost of the exploration phase, and should be bounded. Let $R_{\text{range}} = \max_{s,a} R(s,a) - \min_{s,a} R(s,a)$ and define

$$N_0 = \max\left[ \left(\frac{2u}{R_{\text{range}}}\right)^2, \left(\frac{8u + 16\gamma \cdot R_{\text{range}} \cdot w}{(1-\gamma)^2 \cdot m}\right)^2, \left(\frac{2\gamma \cdot w}{(1-\gamma)^2} \cdot \frac{6\varepsilon + m \cdot \mu_{\min}}{m \cdot \mu_{\min}^2}\right)^2 \right]$$

Then define $k_0$ as

$$k_0 = 8\log(\frac{1}{p}) + \frac{4\alpha^2 N_0}{\mu_{\min}^2} \cdot \frac{\log 16SA + c_2 \cdot (\log \frac{1}{8SA\cdot\beta})}{c_1 \cdot (\log \frac{1}{8SA\cdot\beta})^2} \tag{8}$$

where $c_1 = 0.08$ and $c_2 = 1.34$. We then have the following lemma:

**Lemma 4.3.** *With probability of at least $1 - 2p$,* U2 *uses at most $k_0$ learners as per* (8) *in the exploration phase.*

This bound is based on the guarantee from Lemma 4.1 on the effectiveness of the exploration phase in collecting observations and analysis of how more observations shrink the confidence intervals and reduce the suboptimality of the attack phase. Together with the guarantee of Eq. (6), Lemma 4.3 gives us the upper bound on the total attack cost in Theorem 3.1.

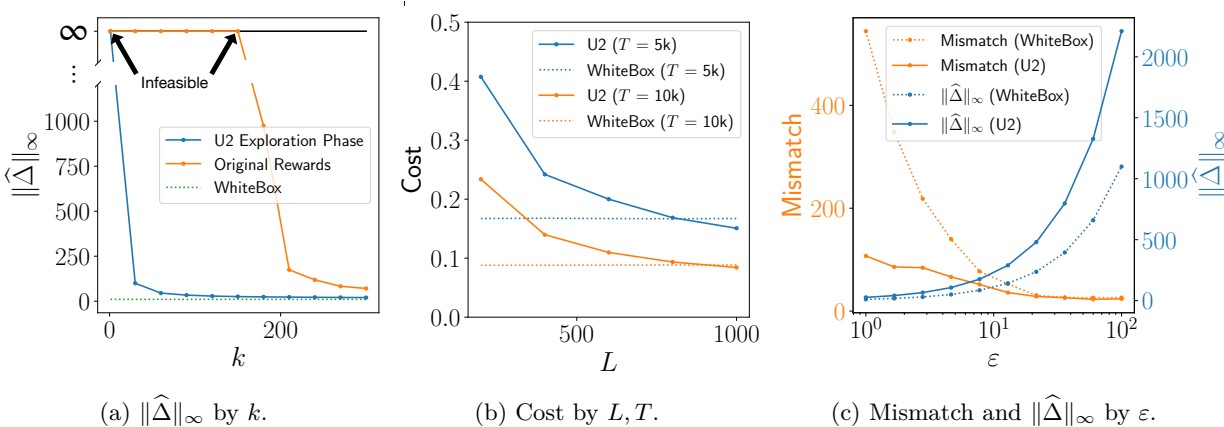

(a) $\|\widehat{\Delta}\|_\infty$ by $k$.      (b) Cost by $L, T$.      (c) Mismatch and $\|\widehat{\Delta}\|_\infty$ by $\varepsilon$.

Figure 1: Results of numerical simulations. Plot **(a)** shows the effectiveness of the exploration phase. Plot **(b)** shows the final cost of the attack for different values of $T, L$. Plot **(c)** demonstrates the difference in the effect of $\varepsilon$ on U2 and white-box attacks. Details are in Section 5.

## 5   Numerical Simulations

We validate our theory by simulating our proposed attack on a grid world environment shown in Fig 2. The environment has $|S| = 3 \times 3$ states, and $|A| = 4$ actions. For each action, there is 0.8 chance to move towards the desired direction. Otherwise, a random other direction is chosen. If a movement is blocked by a wall, the agent stays in place. The agent starts at top left corner. The three actions leading to the middle cell have reward 1 and other rewards are zero. The target policy $\pi_\dagger$ is to keep the agent in the top-right corner and is shown with red arrows. All learners use Q-learning with UCB exploration algorithm (Wang et al., 2020), which is a sample efficient model-free RL algorithm with theoretic guarantee for infinite-horizon MDPs.

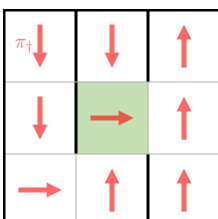

Figure 2: The environment used in the experiments. The red arrows show the target policy.

The first experiment shows how the reward poisoning of U2 in the exploration phase is forcing the learner to explore the environment for the attacker. We evaluate the $\|\widehat{\Delta}\|_\infty$ of the attack phase for various number of learners in the exploration phase. As shown in Fig 1a, the $\|\widehat{\Delta}\|_\infty$ decreases much faster when the rewards are poisoned, which leads to much better guarantee for the cost of the attack. Also note that the attack is not feasible with too little exploration, since the $\mu_{\text{low}}^{\pi_\dagger \{s;a\}}(s)$ term in (4) can be zero when confidence intervals over transitions are too large.

The second experiment shown in Fig 1b evaluates the final value of $\text{COST}(T, L)$ for different values of $T, L$, and $m = 200, \varepsilon = 1$. We observe that just like the upper bound on the cost in Theorem 3.1, the actual cost of U2 decreases with both $T$ and $L$. Even though the guaranteed upper bound on the cost of U2 is always higher than white-box's bound, the actual cost of U2 is in some cases smaller. The reason is that U2 poisons the rewards to create at least an $\varepsilon$ gap between the target policy and other policies for all possible MDPs, but the actual gap created for the true environment is with high probability larger than $\varepsilon$. Due to this difference between the actual gap and $\varepsilon$, U2 is effectively operating similar to the white-box attack with higher $\varepsilon$. This shift is apparent in Fig 1c which shows the trade-off in choice of $\varepsilon$. Larger gap causes the learner to converge to the target policy faster and decreases mismatches and the number of poisonings. On the other hand, it requires larger $\Delta$ and higher per step cost. Fig 1c shows that both of these trends are shifted between U2 and white-box attack, which shows the dependence on $\varepsilon$ is different for these two attack strategies, and for the same $\varepsilon$, U2 can perform even better than the white-box attack.

# 6    Conclusions and Discussion

In this work, we studied the challenging problem of black-box reward poisoning attacks against reinforcement learning, where the adversary starts off having no information on either the environment or the agents' learning algorithm. We proposed an explore-and-exploit style attack U2, that can "hijack" the RL agents to efficiently collect data about the environment, and then carry out a near-optimal attack. We showed that surprisingly, with appropriate choice of hyperparameters, U2 can achieve an attack cost not much worse than the optimal white-box attack. There are potential negative social impact of our algorithms, e.g. utilized by malicious parties. But on the positive side, it also motivates the development of robust RL algorithms to counter these attacks.

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

## Table of Contents

In this section we provide a brief description of the content provided in the appendices of the paper.

- Appendix A provides additional related works.

- Appendix B provides the details on numerical simulations.

- Appendix F provides the proofs for Theorem 3.1.

- Appendix G provides the analysis of Remark 4.2.

## A   Additional Related Works

**No-regret RL algorithms.**   There is a long history of research on no-regret RL algorithms, and in the tabular MDP case, this problem is now considered well-understood. For example, in the episodic setting, the UCRL2 algorithm (Auer et al., 2009) achieves $O(\sqrt{H^4 S^2 AT})$ regret, where $H$ is the episode length, $S$ is the state space size, $A$ is the action space size, and $T$ is the total number of steps. The UCBVI algorithm (Azar et al., 2017; Dann et al., 2017) achieves the optimal $O(\sqrt{H^2 SAT})$ regret matching the lower bound (Osband & Van Roy, 2016; Dann & Brunskill, 2015). More recently, model-free methods (Jin et al., 2018) and policy-based methods (Cai et al., 2020) have all been shown to be able to achieve the same optimal regret bound. In our work, we assume that the learner is implementing a no-regret algorithm, i.e. the regret scales sublinearly with $T$. We will show that the learning efficiency of the learner will "backfire" on itself in presence of an attack.

**Test-time attacks against RL.**   Earlier work on adversarial attacks against RL studied *test-time* attacks, where an adversary aims to manipulate the perceived state of the environment to mislead a fixed and deployed RL policy to perform an incorrect action (Huang et al., 2017; Lin et al., 2017; Kos & Song, 2017; Behzadan & Munir, 2017). For example, in Atari games, the attacker can make small pixel perturbation to a frame, similar to adversarial attacks on image classification  (Goodfellow et al., 2014)), to induce an action $\pi(s_t^\dagger) \neq \pi(s_t)$. Although test-time attacks can severely impact the performance of a deployed and fixed policy $\pi$, they do not modify $\pi$ itself, and thus the adversarial impact will disappear as soon as the attack terminates. On the other hand, poisoning attacks are *training-time* attacks that aim at changing the learned policy and thus have a long-term effect.

**Poisoning attacks and teaching.**   Poisoning attacks is mathematically equivalent to the formulation of machine teaching with the teacher being the adversary (Goldman & Kearns, 1995; Zhu, 2015; Singla et al., 2014; Zhu et al., 2018; Chen et al., 2018; Mansouri et al., 2019; Peltola et al., 2019). A recent line of research has studied robust notions of teaching in settings where the teacher has limited information about the learner's dynamics (Dasgupta et al., 2019; Devidze et al., 2020; Cicalese et al., 2020), however, these works only consider supervised learning settings.

There have been a number of recent works on teaching an RL agent via providing an optimized curriculum of demonstrations (Cakmak & Lopes, 2012; Walsh & Goschin, 2012; Hadfield-Menell et al., 2016; Haug et al., 2018; Kamalaruban et al., 2019; Tschiatschek et al., 2019; Brown & Niekum, 2019). However, most of these works have focused on imitation-learning based RL agents who learn from provided demonstrations without any reward feedback (Osa et al., 2018). Given that we consider RL agents who find policies based on rewards, our work is technically very different from theirs. A recent work of (Zhang et al., 2020a) studies the problem of teaching Q-learning algorithm, however, considers the white-box setting. There is also related literature on changing the behavior of an RL agent via *reward shaping* (Ng et al., 1999; Asmuth et al., 2008); here

the reward function is changed to only speed up the convergence of the learning algorithm while ensuring that the optimal policy in the modified environment is unchanged.

**Reward Shaping:** While this paper is phrased from the adversarial angle, the framework and techniques are also applicable to the *teaching* setting, where a *teacher* aims to guide the agent to learn the *optimal policy* as soon as possible, by designing the reward signal. Traditionally, reward shaping and more specifically potential-based reward shaping (Ng et al., 1999) has been shown able to speed up learning while preserving the optimal policy. (Devlin & Kudenko, 2012) extend potential-based reward shaping to be time-varying while remains policy-preserving. More recently, intrinsic motivations(Schmidhuber, 1991; Oudeyer & Kaplan, 2009; Barto, 2013; Bellemare et al., 2016) was introduced as a new form of reward shaping to encourage exploration and thus speed up learning. Our work contributes by mathematically defining the teaching via reward shaping task as an optimal control problem, and provide computational tools that solve for problem-dependent high-performing reward shaping strategies.

## B    Details of Numerical Simulations

On all experiments, the learner's UCB-Q algorithm is run with hyperparameters $\varepsilon_{\mathrm{UCBQ}} = 0.01, \delta_{\mathrm{UCBQ}} = 0.05$, and to allow convergence in a reasonable rate, the UCB bonuses are divided by 5000. All experiments are run five times and the mean of quantities are reported. We set $\gamma = 0.9$, $\sigma = 2$, $p = 0.05$, and $\lambda = 1$ for all the experiments. In the first experiment, in which the effectiveness of exploration phase is evaluated, we have $T = 5000$, $L = 200$ and $\varepsilon = 1$.In the second experiment, $m = 200, \varepsilon = 1$ is used with various values of $T, L$.

## C    Proof for the white-box attack

**Lemma C.1** (Lemma 3.2). *The optimal solution for* (P1) *is* $\Delta_M^*$. *Moreover,* $\Delta \colon S \times A \to \mathbb{R}$ *is a feasible solution of* (P1) *if and only if for every state $s$ and action $a$,* $\Delta(s,a) \geq \Delta_M^*(s,a)$ *and* $\Delta(s, \pi_\dagger(s)) = 0$.

For completeness, we include the proof of Lemma 3.2 for the white-box attack that immediately follows from the following two lemmas in (Rakhsha et al., 2020b).

**Lemma C.2.** *(Rakhsha et al., 2020b) Policy $\pi$ is $\varepsilon$-robust optimal* iff *we have $\rho^\pi \geq \rho^{\pi\{s;a\}} + \varepsilon$ for every state $s$ and action $a \neq \pi(s)$.*

**Lemma C.3.** *(Schulman et al., 2015) For two policies $\pi$ and $\pi'$ we have:*

$$\rho_M^\pi - \rho_M^{\pi'} = \sum_{s \in \mathcal{S}} \mu_M^{\pi'}(s) \big( Q_M^\pi(s, \pi(s)) - Q_M^\pi(s, \pi'(s)) \big). \tag{9}$$

**Proof of Lemma C.1.** Based on these two lemmas, we note that $\Delta$ is a feasible solution for (P1) if and only if for every $s, a$

$$\varepsilon \leq \rho_{M'}^{\pi_\dagger} - \rho_{M'}^{\pi_\dagger \{s;a\}} = \mu_{M'}^{\pi_\dagger \{s;a\}}(s)\big(V_{M'}^{\pi_\dagger}(s) - Q_{M'}^{\pi_\dagger}(s,a)\big) = \mu_M^{\pi_\dagger \{s;a\}}(s)\big(V_M^{\pi}(s) - Q_M^{\pi_\dagger}(s,a) + \Delta(s,a)\big) \tag{10}$$

where $M' = (S, A, R - \Delta, P)$. Rearranging this inequality gives us the condition in the lemma:

$$\Delta(s,a) \geq Q_M^{\pi_\dagger}(s,a) - V_M^\pi(s) \frac{\varepsilon}{\mu_M^{\pi_\dagger \{s;a\}}(s)} = \Delta_M^*(s,a) \tag{11}$$

∎

## D    Proofs for the exploration phase

In this section, we first prove Lemma 4.1 which lower-bounds the number of visits to each $(s,a)$ pair by each learner in the exploration phase. Then we give the details of Lemma 4.2.

**Lemma D.1** (Lemma 4.1). *Let $s, a$ be an arbitrary state and action pair, and $g(s, a) = \min_{\pi:\pi(s)=a} \mu_M^\pi(s)$.*
*Assume $4\beta \leq \delta$ and $\frac{\alpha}{g(s,a)} < \frac{1}{2\sqrt{2}}$. If the feedback as in (2) is given to a learner, then at the end of $T$ steps,*
*with probability of at least $1 - \delta$, the action $a$ is chosen from state $s$ for at least*

$$\frac{g(s, a)^2}{\alpha^2} \cdot \frac{c_1 \cdot (\log \frac{\delta}{4\beta})^2}{\log \frac{8}{\delta} + c_2 \cdot (\log \frac{\delta}{4\beta})} \tag{12}$$

*number of times, where $c_1 = 0.08$ and $c_2 = 1.34$.*

**Proof of Lemma D.1.** We will show that with probability of at least $1 - \delta$, we have

$$N(s, a) \geq t^* := \frac{g(s, a)^2}{\alpha^2} \cdot \frac{c_1 \cdot (\log \delta/4\beta)^2}{\log \frac{8}{\delta} + c_2 \cdot (\log \delta/4\beta)}. \tag{13}$$

.

Define the probability distribution $\mathcal{B}(p)$ with parameter $p$ as the following

$$\mathbb{P}[X = x | X \sim \mathcal{B}(p)] = \begin{cases} p & \text{if } x = 1 \\ 1 - p & \text{if } x = -1 \end{cases} \tag{14}$$

We consider three possible reward distributions during the exploration phase. We call these possibilities "*hypotheses*" $H_0, H_1$ and $H_2$. $H_0$ is the actual reward distribution simulated for the learner:

$$H_0 : \quad r_t' \sim \mathcal{B}(\frac{1}{2}) \tag{15}$$

$H_1$ is an alternative reward distributions in which $s, a$ is taken in all $\alpha$-optimal policies, and $H_2$, in contrary, is a reward distributions in which $s, a$ is not taken in any $\alpha$-optimal policies:

$$H_1 : \quad r_t' \sim \begin{cases} \mathcal{B}(\frac{1}{2} + \frac{\alpha}{2g(s,a)}) & \text{if } s_t = s, a_t = a \\ \mathcal{B}(\frac{1}{2}) & \text{otherwise} \end{cases} \tag{16}$$

$$H_2 : \quad r_t' \sim \begin{cases} \mathcal{B}(\frac{1}{2} - \frac{\alpha}{2g(s,a)}) & \text{if } s_t = s, a_t = a \\ \mathcal{B}(\frac{1}{2}) & \text{otherwise} \end{cases} \tag{17}$$

Let $M_E^+ = (S, A, R_E^+, P)$ be the MDP with rewards described in $H_1$ and Let $M_E^- = (S, A, R_E^-, P)$ be the MDP with rewards described in $H_2$. The following lemma formalizes this construction.

**Lemma D.2.** *For every $\alpha$-optimal policy $\pi$ in $M_E^+$, we have $\pi(s) = a$. In contrast, for every $\alpha$-optimal policy $\pi$ in $M_E^-$, we have $\pi(s) \neq a$.*

*Proof.* Let $\pi$ and $\pi'$ be arbitrary policies such that $\pi(s) \neq a$ and $\pi'(s) = a$. We have $\rho_{M_E^+}^\pi = \rho_{M_E^-}^\pi = 0$. We can also write

$$\rho_{M_E^+}^{\pi'} = \sum_{s' \in S} \mu_{M_E^+}^{\pi'}(s') R_E^+(s', \pi'(s')) = \frac{\alpha}{g(s,a)} \mu_M^{\pi'}(s) \geq \alpha \tag{18}$$

Thus, $\rho_{M_E^+}^\pi \leq \rho_{M_E^+}^{\pi'} - \alpha$, which shows $\pi$ is not $\alpha$-optimal and proves the first part. For the second part, we write

$$\rho_{M_E^-}^{\pi'} = \sum_{s' \in S} \mu_{M_E^-}^{\pi'}(s') R_E^-(s', \pi'(s')) = -\frac{\alpha}{g(s,a)} \mu_M^{\pi'}(s) \leq -\alpha \tag{19}$$

Consequently, $\rho_{M_E^+}^{\pi'} \leq \rho_{M_E^+}^\pi - \alpha$, and therefore $\pi'$ is not $\alpha$-optimal. ∎

Next, we use the same argument as in the classic lower bound construction in stochastic bandits (Mannor & Tsitsiklis, 2004). The proof is based on the following idea: a sequence of observations in which $(s, a)$ is rarely visited has similar likelihood under all $H_0$, $H_1$, and $H_2$. Thus, if these sequences appear with high probability under $H_0$, they also will happen with high probability under both $H_1$, and $H_2$ too. If in the majority of these sequences, the learner decides to pick $a$ in $s$, it will with a large probability of incur large optimality gap in $H_2$. On the other hand, if the learner decides not to pick $a$ in $s$, will with a large probability of incur large optimality gap in $H_2$.

First, we define some events that are used in our analysis. Let $A$ be the event that the bound is not true, i.e.

$$A = \{N(s, a) < t^*\}. \tag{20}$$

Next, let $\pi^*$ be the final chosen policy by the learner and $B$ denote the event in which $(s, a)$ is chosen by $\pi^*$. More specifically,

$$B = \{\pi^*(s) = a\}. \tag{21}$$

Finally let $K_n = \sum_{i=1}^{n} \mathbb{1}\left[r'_{t_i(s,a)} = 1\right]$ where $t_i(s, a)$ is the $i$-th time step when $s, a$ is chosen and define event $C$ as the following:

$$C = \left\{ \max_{1 \leq n \leq t^*} |2K_n - n| \leq \sqrt{\frac{8t^*}{3} \log \frac{8}{\delta}} \right\} \tag{22}$$

Let $P_0, P_1$ and $P_2$ denote the probability functions under $H_0$, $H_1$, and $H_2$, respectively. We will show that if $P_0(A) \geq \delta$, either $P_1(B^c) \geq \beta$ or $P_2(B) \geq \beta$ where $B^c$ is the complement of $B$. Based on Lemma D.2, this contradicts the learner's guarantee and will prove the lemma. Now, assume that $P_0(A) \geq \delta$. Then, either $P_0(A \cap B) \geq \frac{\delta}{2}$ or $P_0(A \cap B^c) \geq \frac{\delta}{2}$. We consider each of these cases separately. Before that, we need to show some intermediate lemmas.

**Lemma D.3.** *We have $P_0(C) \geq 1 - \frac{\delta}{4}$.*

*Proof.* Let $p = \sqrt{\frac{8t^*}{3} \log \frac{8}{\delta}}$. The result is trivial if $p \geq t^*$. For $p \leq t^*$, from the maximal Bernstein inequality (Theorem B.2 in (Rio, 2017))

$$P_0 \left( \max_{1 \leq n \leq t^*} (2K_n - n) > p \right) \leq \exp\left( -\frac{p^2}{2t^* + 2p/3} \right) \tag{23}$$

$$\leq \exp\left( -\frac{p^2}{2t^* + 2t^*/3} \right) \tag{24}$$

$$= \delta/8 \tag{25}$$

Similarly, we have

$$P_0 \left( \max_{1 \leq n \leq t^*} (n - 2K_n) > p \right) \leq \delta/8 \tag{26}$$

From the union bound, we get

$$P_0 \left( \max_{1 \leq n \leq t^*} |2K_n - n| > p \right) \leq \delta/4 \tag{27}$$

It means that $P_0(C) > 1 - \delta/4$. ∎

**Lemma D.4.** *(Mannor & Tsitsiklis, 2004) If $0 \leq x \leq 1/\sqrt{2}$, then $\log(1 - x) \geq -dx$ where $d = 1.78$.*

Let $Y$ be the sequence of all the chosen actions, received rewards, states visited during the whole interaction with the learner, and the final chosen policy by the learner. We define the likelihood functions $L_0, L_1$ and $L_2$ for each of the hypotheses:

$$L_i(y) = P_i(Y = y) \tag{28}$$

The following lemmas show a lower bound on likelihood of observed history if $A$ happens.

**Lemma D.5.** *If $y \in A \cap C$, then $\frac{L_1(y)}{L_0(y)} \geq \frac{4\beta}{\delta}$.*

*Proof.* Note that the transition probabilities are the same under $H_0$ and $H_1$. Also, conditioned on the history up to time $t$, the choice of action $a_t$ has the same likelihood under $H_0$ and $H_1$ as it only depends on the learner's internal stochasticity. Finally, the received reward has the same distribution unless $s_t = s, a_t = a$. For brevity, let $N = N(s, a), K = K_N, \varepsilon = \frac{\alpha}{2g(s,a)}$. We can write

$$\frac{L_1(y)}{L_0(y)} = \frac{(\frac{1}{2} + \varepsilon)^K \cdot (\frac{1}{2} - \varepsilon)^{N-K}}{(\frac{1}{2})^N} \tag{29}$$

$$= (1 + 2\varepsilon)^K \cdot (1 - 2\varepsilon)^{N-K} \tag{30}$$

$$= (1 - 4\varepsilon^2)^K \cdot (1 - 2\varepsilon)^{N-2K} \tag{31}$$

Since $A \cap C$ has happened, we have

$$K \leq N < t^* \tag{32}$$

$$N - 2K \leq \sqrt{\frac{8t^*}{3} \log \frac{8}{\delta}} \tag{33}$$

Thus using Lemma D.4, we can write:

$$\log \frac{L_1(y)}{L_0(y)} = K \cdot \log(1 - 4\varepsilon^2) + (N - 2K) \cdot \log(1 - 2\varepsilon) \tag{34}$$

$$\geq -t^* \cdot 4d\varepsilon^2 - \sqrt{\frac{8t^*}{3} \log \frac{8}{\delta}} \cdot 2d\varepsilon \tag{35}$$

Thus, we need to show

$$t^* \cdot 4d\varepsilon^2 + \sqrt{\frac{8t^*}{3} \log \frac{8}{\delta}} \cdot 2d\varepsilon \leq \log \frac{\delta}{4\beta} \tag{36}$$

Now let $t^* = \frac{1}{\varepsilon^2} \cdot z^2$, $b = \sqrt{\frac{2}{3} \log \frac{8}{\delta}}$, and $c = \frac{1}{4d} \log \frac{\delta}{4\beta}$. Then, (36) can be rewritten as

$$z^2 + bz - c \leq 0 \tag{37}$$

Since $\delta \geq 4\beta$, we have $c > 0$, and therefore, this is true if $z \geq 0$ and

$$z \leq \frac{-b + \sqrt{b^2 + 4c}}{2} = \frac{2c}{\sqrt{b^2 + 4c} + b} \tag{38}$$

Thus, by Jensen's inequality, it satisfies to have $z \leq \frac{c}{\sqrt{b^2 + 2c}}$ which means we need to have:

$$t^* = \frac{1}{\varepsilon^2} \cdot z^2 \leq \frac{1}{\varepsilon^2} \frac{\frac{1}{16d^2} (\log \frac{\delta}{4\beta})^2}{\frac{2}{3} \log \frac{8}{\delta} + \frac{1}{2d} \log \frac{\delta}{4\beta}} \tag{39}$$

$$= \frac{4g(s, a)^2}{\alpha^2} \frac{\frac{3}{32d^2} \cdot (\log \frac{\delta}{4\beta})^2}{\log \frac{8}{\delta} + \frac{3}{4d} \log \frac{\delta}{4\beta}} \tag{40}$$

which is true when $c_1 \leq \frac{3}{8d^2}$ and $c_2 \geq \frac{3}{4d}$. $\blacksquare$

**Lemma D.6.** *If $y \in A \cap C$, then $\frac{L_2(y)}{L_0(y)} \geq \frac{4\beta}{\delta}$.*

*Proof.* Following the same argument in proof of Lemma.D.5, we can write

$$\frac{L_2(y)}{L_0(y)} = \frac{(\frac{1}{2} - \varepsilon)^K \cdot (\frac{1}{2} + \varepsilon)^{N-K}}{(\frac{1}{2})^N} \tag{41}$$

$$= (1 - 2\varepsilon)^K \cdot (1 + 2\varepsilon)^{N-K} \tag{42}$$

$$= (1 - 4\varepsilon^2)^K \cdot (1 + 2\varepsilon)^{N-2K} \tag{43}$$

where $N = N(s,a), K = K_N, \varepsilon = \frac{\alpha}{g(s,a)}$. Since $A \cap C$ has happened, we have

$$K \leq N < t^* \tag{44}$$

$$N - 2K \geq -\sqrt{\frac{8t^*}{3} \log \frac{8}{\delta}} \tag{45}$$

Thus using Lemma D.4, we can write:

$$\log \frac{L_1(y)}{L_0(y)} = K \cdot \log(1 - 4\varepsilon^2) + (N - 2K) \cdot \log(1 + 2\varepsilon) \tag{46}$$

$$\geq t^* \cdot \log(1 - 4\varepsilon^2) - \sqrt{\frac{8t^*}{3} \log \frac{8}{\delta}} \cdot \log(1 + 2\varepsilon) \tag{47}$$

$$\geq -t^* \cdot 4d\varepsilon^2 - \sqrt{\frac{8t^*}{3} \log \frac{8}{\delta}} \cdot 2\varepsilon \tag{48}$$

$$\tag{49}$$

where we used $\log(1 + x) \leq x$ for $x \geq 0$. Thus, we need to show

$$t^* \cdot 4d\varepsilon^2 + \sqrt{\frac{8t^*}{3} \log \frac{8}{\delta}} \cdot 2\varepsilon \leq \log \frac{\delta}{4\beta} \tag{50}$$

Now let $t^* = \frac{1}{\varepsilon^2} \cdot z^2$, $b = \sqrt{\frac{2}{3} \log \frac{8}{\delta}}$, and $c = \frac{1}{4} \log \frac{\delta}{4\beta}$. Then (50) can be rewritten as

$$dz^2 + bz - c \geq 0 \tag{51}$$

Since $\delta \geq 4\beta$, we have $c > 0$, and therefore, this is true if $z \geq 0$ and

$$z \leq \frac{-b + \sqrt{b^2 + 4cd}}{2d} = \frac{2c}{\sqrt{b^2 + 4cd} + b} \tag{52}$$

Thus, by Jensen's inequality, it satisfies to have $z \leq \frac{c}{\sqrt{b^2 + 2cd}}$ which means we need to have:

$$t^* = \frac{1}{\varepsilon^2} \cdot z^2 \leq \frac{1}{\varepsilon^2} \frac{\frac{1}{16}(\log \frac{\delta}{4\beta})^2}{\frac{2}{3} \log \frac{8}{\delta} + \frac{d}{2} \log \frac{\delta}{4\beta}} \tag{53}$$

$$= \frac{4g(s,a)^2}{\alpha^2} \frac{\frac{3}{32} \cdot (\log \frac{\delta}{4\beta})^2}{\log \frac{8}{\delta} + \frac{3d}{4} \cdot \log \frac{\delta}{4\beta}} \tag{54}$$

which is true when $c_1 \leq \frac{3}{8}$ and $c_2 \geq 3d/4$. ∎

Finally, to prove Lemma 4.1, assume $P_0(A) \geq \delta$ and consider two cases: $P_0(A \cap B^c) \geq \frac{\delta}{2}$ and $P_0(A \cap B) \geq \frac{\delta}{2}$. If $P_0(A \cap B^c) \geq \frac{\delta}{2}$, first note that $P_0(A \cap B^c \cap C) \geq \delta/4$. Now we write

$$P_1(B^c) \geq P_1(A \cap B^c \cap C) \tag{55}$$

$$= E_1[\mathbf{1}_{A \cap B^c \cap C}] \tag{56}$$

$$= E_0[\mathbf{1}_{A \cap B^c \cap C} \frac{L_1(Y)}{L_0(Y)}] \tag{57}$$

$$\geq E_0[\mathbf{1}_{A \cap B^c \cap C} \frac{4\beta}{\delta}] \tag{58}$$

$$= \frac{4\beta}{\delta} \cdot P_0(A \cap B^c \cap C) \tag{59}$$

$$\geq \beta \tag{60}$$

where $\mathbf{1}_X$ denotes the indicator function of event $X$ and we used Lemma D.5.

Similarly, when $P_0(A \cap B) \geq \frac{\delta}{2}$, we have $P_0(A \cap B \cap C) \geq \delta/4$, and we write

$$P_2(B) \geq P_2(A \cap B \cap C) \tag{61}$$

$$= E_2[\mathbf{1}_{A \cap B \cap C}] \tag{62}$$

$$= E_0[\mathbf{1}_{A \cap B \cap C} \frac{L_2(Y)}{L_0(Y)}] \tag{63}$$

$$\geq E_0[\mathbf{1}_{A \cap B \cap C} \frac{4\beta}{\delta}] \tag{64}$$

$$= \frac{4\beta}{\delta} \cdot P_0(A \cap B \cap C) \tag{65}$$

$$\geq \beta \tag{66}$$

In both cases, the assumption on learner's performance in wrong. This shows that $P_0(A) < \delta$. ∎

**Lemma D.7** (Lemma 4.2). *With probability of at least $1 - p/L$, $M \in \mathcal{M}$.*

**Proof of Lemma D.7.** From Hoeffding's inequality, for every $s, a$ we have

$$\mathbb{P}\left(|R(s,a) - \widehat{R}(s,a)| > z\right) \leq 2\exp\left(\frac{-N(s,a)z^2}{2\sigma^2}\right) \tag{67}$$

Letting $z = \frac{u}{\sqrt{N_{\min}}}$, we get

$$\mathbb{P}\left(|R(s,a) - \widehat{R}(s,a)| > \frac{u}{\sqrt{N_{\min}}}\right) \leq 2\exp\left(\frac{-N(s,a)u^2}{2\sigma^2 \cdot N_{\min}}\right) \tag{68}$$

$$\leq \frac{p}{2SAL} \tag{69}$$

Also, (Weissman et al., 2003) bounds the $\ell_1$ distance between empirical distribution $\widehat{d}$ with $N$ samples and true distribution $d$ over $n$ outcomes as

$$\mathbb{P}\left(\|d - \widehat{d}\|_1 \geq z\right) \leq (2^n - 2)\exp\left(-\frac{Nz^2}{2}\right) \tag{70}$$

Thus, for every $s, a$ we have

$$\mathbb{P}\left(\|P(s,a,.) - \widehat{P}(s,a,.)\|_1 \geq \frac{w}{\sqrt{N_{\min}}}\right) \leq (2^S - 2)\exp\left(-\frac{N(s,a)w^2}{2N_{\min}}\right) \tag{71}$$

$$\leq 2^S \exp\left(-\frac{w^2}{2}\right) \tag{72}$$

$$= \frac{p}{2SAL} \tag{73}$$

The lemma immediately follows from the union bound and above bounds. ∎

The following result is an immediate consequence of Lemma D.7.

**Corollary D.1.** *With probability of at least $1 - p$, we have $M \in \mathcal{M}$ for all $\mathcal{M}$s built after each of the learners in the exploration phase.*

## E   Proof of Stopping Condition

In this section, we provide rigorous proofs on various consequences induced by the stopping condition (7):

$$2\widehat{e}_Q + \frac{\varepsilon}{\mu_{\text{low}}^{\pi_\dagger\{s;a\}}(s)} - \frac{\varepsilon}{\mu_{\text{low}}^{\pi_\dagger\{s;a\}}(s) + e_\mu} \leq m \tag{74}$$

In what follows, define event $B$ as

$$B := \{M \in \mathcal{M} \text{ for all } \mathcal{M}\text{s built after each of the learners in the exploration phase}\} \tag{75}$$

From Corollary D.1 we have $\mathbb{P}(B) \geq 1 - p$.

We make use of the *simulation lemma*:

**Lemma E.1.** *(Simulation Lemma (Strehl & Littman, 2008))* Let $M_1 = (S, A, R_1, P_1, \gamma)$ and $M_2 = (S, A, R_2, P_2, \gamma)$ *be two MDPs with reward range $R_{range}$. The following condition holds for all states $s$, actions $a$, and stationary, deterministic policies $\pi$:*

$$|Q_1^\pi(s, a) - Q_2^\pi(s, a)| \leq \frac{1}{(1-\gamma)^2}(\|R_1 - R_2\|_\infty + \gamma(R_{range})\|P_1 - P_2\|_\infty) \tag{76}$$

Our first result says that the stopping condition guarantees that the attack cost is at most $m$ worse than the white-box attack:

**Lemma E.2.** *Under event $B$, the stopping condition* (74) *guarantees that $\widehat{\Delta}(s, a) \leq \Delta_M^*(s, a) + m$ for every $s, a$.*

**Proof of Lemma E.2.** Lemma E.1 and event $B$ imply

$$Q_M^{\pi_\dagger}(s, a) \geq Q_{\text{high}}^{\pi_\dagger}(s, a) - e_Q \tag{77}$$

$$V_M^{\pi_\dagger}(s) \leq V_{\text{low}}^{\pi_\dagger}(s) + e_Q \tag{78}$$

where again

$$e_Q = \frac{2u + 2\gamma \cdot R_{\text{range}} \cdot w}{(1 - \gamma)^2 \cdot \sqrt{N_{\min}}} \ , \ e_\mu = \frac{2\gamma \cdot w}{(1 - \gamma)^2 \cdot \sqrt{N_{\min}}}. \tag{79}$$

Let $\overline{M} = \arg\min_{\widetilde{M} \in \mathcal{M}} \mu_{\widetilde{M}}^{\pi_\dagger\{s;a\}}(s)$, which means $\mu_{\text{low}}^{\pi_\dagger\{s;a\}} = \mu_{\overline{M}}^{\pi_\dagger\{s;a\}}$. In $M_s$ and $\overline{M}_s$, all the rewards are the same and either 0 or 1. Thus, utilizing the simulation lemma,

$$\mu_M^{\pi_\dagger\{s;a\}}(s) = V_{M_s}^{\pi_\dagger\{s;a\}}(s^0) \tag{80}$$

$$\leq V_{\overline{M}}^{\pi_\dagger\{s;a\}}(s^0) + \frac{2\gamma w}{(1-\gamma)^2 \cdot \sqrt{N_{\min}}} \tag{81}$$

$$= \mu_{\text{low}}^{\pi_\dagger\{s;a\}}(s) + e_\mu \tag{82}$$

Under event $B$, all the confidence intervals hold and therefore $\widehat{R}_{\text{range}} \geq R_{\text{range}}$. In that case, from Lemma E.1, we have:

$$V_M^{\pi\dagger}(s) \leq V_{\text{low}}^{\pi\dagger}(s) + \widehat{e}_Q \tag{83}$$

$$Q_M^{\pi\dagger}(s,a) \geq Q_{\text{high}}^{\pi\dagger}(s,a) - \widehat{e}_Q \tag{84}$$

Thus, once (7) is satisfied, we have

$$m + Q_M^{\pi\dagger}(s,a) - V_M^{\pi\dagger}(s) + \frac{\varepsilon}{\mu_M^{\pi\dagger\{s;a\}}(s)} \tag{85}$$

$$\geq m + Q_{\text{high}}^{\pi\dagger}(s,a) - V_{\text{low}}^{\pi\dagger}(s) - 2\widehat{e}_Q + \frac{\varepsilon}{\mu_{\text{low}}^{\pi\dagger\{s;a\}}(s) + e_\mu} \tag{86}$$

$$\geq 2\widehat{e}_Q + \frac{\varepsilon}{\mu_{\text{low}}^{\pi\dagger\{s;a\}}(s)} - \frac{\varepsilon}{\mu_{\text{low}}^{\pi\dagger\{s;a\}}(s) + e_\mu} + Q_{\text{high}}^{\pi\dagger}(s,a) - V_{\text{low}}^{\pi\dagger}(s) - 2\widehat{e}_Q + \frac{\varepsilon}{\mu_{\text{low}}^{\pi\dagger\{s;a\}}(s) + e_\mu} \tag{87}$$

$$= Q_{\text{high}}^{\pi\dagger}(s,a) - V_{\text{low}}^{\pi\dagger}(s) + \frac{\varepsilon}{\mu_{\text{low}}^{\pi\dagger\{s;a\}}(s)} \tag{88}$$

Consequently,

$$m \geq \left( Q_{\text{high}}^{\pi\dagger}(s,a) - V_{\text{low}}^{\pi\dagger}(s) + \frac{\varepsilon}{\mu_{\text{low}}^{\pi\dagger\{s;a\}}(s)} \right) - \left( Q_M^{\pi\dagger}(s,a) - V_M^{\pi\dagger}(s) + \frac{\varepsilon}{\mu_M^{\pi\dagger\{s;a\}}(s)} \right) \tag{89}$$

$$\geq \left[ Q_{\text{high}}^{\pi\dagger}(s,a) - V_{\text{low}}^{\pi\dagger}(s) + \frac{\varepsilon}{\mu_{\text{low}}^{\pi\dagger\{s;a\}}(s)} \right]^+ - \left[ Q_M^{\pi\dagger}(s,a) - V_M^{\pi\dagger}(s) + \frac{\varepsilon}{\mu_M^{\pi\dagger\{s;a\}}(s)} \right]^+ \tag{90}$$

$$= \widehat{\Delta}(s,a) - \Delta_M^*(s,a) \tag{91}$$

∎

**Lemma E.3.** *Under event $B$, the stopping condition is satisfied after at most $N_0$ observations of each $(s,a)$ pair.*

**Proof of Lemma E.3.** We show that after $N_0$ observations

$$2\widehat{e}_Q \leq \frac{m}{2} \tag{92}$$

$$\frac{\varepsilon}{\mu_{\text{low}}^{\pi\dagger\{s;a\}}(s)} - \frac{\varepsilon}{\mu_{\text{low}}^{\pi\dagger\{s;a\}}(s) + e_\mu} \leq \frac{m}{2} \tag{93}$$

For the first part, note that $N_0 \geq \left( \frac{2u}{R_{\text{range}}} \right)^2$. Thus, under event $B$, we have

$$\widehat{R}_{\text{range}} \leq R_{\text{range}} + \frac{2u}{\sqrt{N_{\min}}} \leq 2R_{\text{range}} \tag{94}$$

Thus,

$$\widehat{e}_Q = \frac{2u + 2\gamma \cdot \widehat{R}_{\text{range}} \cdot w}{(1-\gamma)^2 \cdot \sqrt{N_{\min}}} \leq \frac{2u + 4\gamma \cdot R_{\text{range}} \cdot w}{(1-\gamma)^2 \cdot \sqrt{N_{\min}}} \tag{95}$$

From $N_{\min} \geq \left( \frac{8u + 16\gamma \cdot R_{\text{range}} \cdot w}{(1-\gamma)^2 \cdot m} \right)^2$ we get $\widehat{e}_Q \leq \frac{m}{4}$.

From

$$N_{\min} \geq \left( \frac{2\gamma \cdot w}{(1-\gamma)^2} \cdot \frac{6\varepsilon + m \cdot \mu_{\min}}{m \cdot \mu_{\min}^2} \right)^2, \tag{96}$$

we get

$$e_\mu = \frac{2\gamma \cdot w}{(1-\gamma)^2 \cdot \sqrt{N_{\min}}} \leq \frac{m \cdot \mu_{\min}^2}{6\varepsilon + m \cdot \mu_{\min}} \tag{97}$$

With the same argument as in (82), we have

$$\mu_M^{\pi_\dagger\{s;a\}}(s) - e_\mu \leq \mu_{\text{low}}^{\pi_\dagger\{s;a\}}(s) \leq \mu_M^{\pi_\dagger\{s;a\}}(s) + e_\mu \tag{98}$$

note that (97) shows that $e_\mu < \mu_{\min} \leq \mu_M^{\pi_\dagger\{s;a\}}(s)$ so we can write

$$\frac{\varepsilon}{\mu_{\text{low}}^{\pi_\dagger\{s;a\}}(s)} - \frac{\varepsilon}{\mu_{\text{low}}^{\pi_\dagger\{s;a\}}(s) + e_\mu} \leq \frac{\varepsilon}{\mu_M^{\pi_\dagger\{s;a\}}(s) - e_\mu} - \frac{\varepsilon}{\mu_M^{\pi_\dagger\{s;a\}}(s) + 2e_\mu} \tag{99}$$

$$= \frac{3\varepsilon e_\mu}{(\mu_M^{\pi_\dagger\{s;a\}}(s) - e_\mu)(\mu_M^{\pi_\dagger\{s;a\}}(s) + 2e_\mu)} \tag{100}$$

$$\leq \frac{3\varepsilon e_\mu}{(\mu_M^{\pi_\dagger\{s;a\}}(s) - e_\mu) \cdot \mu_M^{\pi_\dagger\{s;a\}}(s)} \tag{101}$$

Thus, it suffices to show

$$\frac{3\varepsilon e_\mu}{(\mu_M^{\pi_\dagger\{s;a\}}(s) - e_\mu) \cdot \mu_M^{\pi_\dagger\{s;a\}}(s)} \leq \frac{m}{2} \tag{102}$$

$$\Leftrightarrow 6\varepsilon e_\mu \leq m \cdot \mu_M^{\pi_\dagger\{s;a\}}(s) \cdot (\mu_M^{\pi_\dagger\{s;a\}}(s) - e_\mu) \tag{103}$$

$$\Leftrightarrow (6\varepsilon + m \cdot \mu_M^{\pi_\dagger\{s;a\}}(s))e_\mu \leq m \cdot \left(\mu_M^{\pi_\dagger\{s;a\}}(s)\right)^2 \tag{104}$$

$$\Leftrightarrow e_\mu \leq \frac{m \cdot \left(\mu_M^{\pi_\dagger\{s;a\}}(s)\right)^2}{6\varepsilon + m \cdot \mu_M^{\pi_\dagger\{s;a\}}(s)} \tag{105}$$

which follows from (97) noting that $\mu_{\min} \leq \mu_M^{\pi_\dagger\{s;a\}}(s)$ and $f(x) = \frac{x}{a+x}$ is increasing for $a, x > 0$. ∎

**Lemma E.4.** *With probability $1 - p$, $N_0$ observations on each $(s, a)$ pair can be made after at most $k_0$ learners.*

**Proof of Lemma E.4.** Setting $\delta = \frac{1}{2SA}$, Lemma 4.1 and union bound imply that with probability of at least $1/2$, each learner give the following number of observations for each $s, a$

$$\frac{\mu_{\min}^2}{\alpha^2} \cdot \frac{c_1 \cdot (\log \frac{1}{8SA \cdot \beta})^2}{\log 16SA + c_2 \cdot (\log \frac{1}{8SA \cdot \beta})} \tag{106}$$

Let $k_1$ be the number of learners among the $k_0$ learners for which the above bound holds. Then after $k_0$ learners, we have at least

$$k_1 \cdot \frac{\mu_{\min}^2}{\alpha^2} \cdot \frac{c_1 \cdot (\log \frac{1}{8SA \cdot \beta})^2}{\log 16SA + c_2 \cdot (\log \frac{1}{8SA \cdot \beta})} \tag{107}$$

observations.

From Hoeffding's inequality we have

$$\mathbb{P}\left[k_1 \leq \left(\frac{1}{2} - \sqrt{\frac{\log 1/p}{2k_0}}\right) \cdot k_0\right] \leq p \tag{108}$$

Thus, with probability of at least $1 - p$,

$$k_1 \geq \left( \frac{1}{2} - \sqrt{\frac{\log 1/p}{2k_0}} \right) \cdot k_0 \tag{109}$$

and consequently

$$N_{\min} \geq \left( \frac{1}{2} - \sqrt{\frac{\log 1/p}{2k_0}} \right) \cdot k_0 \cdot \frac{\mu_{\min}^2}{\alpha^2} \cdot \frac{c_1 \cdot (\log \frac{1}{8SA \cdot \beta})^2}{\log 16SA + c_2 \cdot (\log \frac{1}{8SA \cdot \beta})} \tag{110}$$

Now we have $k_0 \geq 8 \log 1/p$ which gives

$$\frac{1}{2} - \sqrt{\frac{\log 1/p}{2k_0}} \geq 1/4 \tag{111}$$

thus

$$N_{\min} \geq \frac{1}{4} \cdot k_0 \cdot \frac{\mu_{\min}^2}{\alpha^2} \cdot \frac{c_1 \cdot (\log \frac{1}{8SA \cdot \beta})^2}{\log 16SA + c_2 \cdot (\log \frac{1}{8SA \cdot \beta})} \geq N_0 \tag{112}$$

∎

## F  Proof of the main theorem

Finally, we prove our main theorem by combining all the building blocks above.

**Theorem F.1** (Theorem 3.1)**.** *For any $m > 0$ and $p \in (0, 1)$, assume that $\alpha < \frac{\mu_{min}}{2\sqrt{2}}$ and $\beta < \frac{1}{8SA}$, then, with probability of at least $1 - 4p$, the cost of* U2 *is bounded by*

$$\text{COST}(T, L) \leq \frac{k_0}{L} \cdot \left( \|R\|_\infty + \sigma \sqrt{2 \log \frac{2k_0 T}{p}} + 1 + \lambda \right) \tag{113}$$
$$+ (\|\Delta_M^*\|_\infty + \lambda + m) \cdot \frac{\text{SUBOPT}(T, \varepsilon, \frac{p}{L})}{T}$$

*where $k_0$ is a function of MDP $M$, $p$, $\alpha$, $\beta$, $\lambda$, $m$, $\varepsilon$, and $L$ as defined in* (8).

**Proof of Theorem F.1.** Let $k_1$ denote the number of learners in the exploration phase, and $\xi_t^{(l)}$ be the noise in reward of step $t$ of learner $l$, i.e. $r_t^{(l)} = R(s_t^{(l)}, a_t^{(l)}) + \xi_t^{(l)}$. For the total cost we have

$$\text{COST}(T, L) = \frac{1}{L \cdot T} \sum_{l=1}^{L} \sum_{t=1}^{T} \left( |r_t^{(l)} - r_t'^{(l)}| + \lambda \mathbb{1} \left[ a_t^{(l)} \neq \pi_\dagger(s_t^{(l)}) \right] \right) \tag{114}$$

$$= \frac{1}{L \cdot T} \sum_{l=1}^{k_1} \sum_{t=1}^{T} \left( |R(s_t^{(l)}, a_t^{(l)}) + \xi_t^{(l)} - r_t'^{(l)}| + \lambda \mathbb{1} \left[ a_t^{(l)} \neq \pi_\dagger(s_t^{(l)}) \right] \right) + \tag{115}$$

$$\frac{1}{L \cdot T} \sum_{l=k_1+1}^{L} \sum_{t=1}^{T} \left( \widehat{\Delta}(s_t^{(l)}, a_t^{(l)}) + \lambda \mathbb{1} \left[ a_t^{(l)} \neq \pi_\dagger(s_t^{(l)}) \right] \right)$$

$$\leq \frac{1}{L \cdot T} \sum_{l=1}^{k_1} \sum_{t=1}^{T} \left( |R(s_t^{(l)}, a_t^{(l)})| + |\xi_t^{(l)}| + 1 + \lambda \right) + \tag{116}$$

$$\frac{1}{L \cdot T} \sum_{l=k_1+1}^{L} \sum_{t=1}^{T} \left( (\widehat{\Delta}(s_t^{(l)}, a_t^{(l)}) + \lambda) \cdot \mathbb{1} \left[ a_t^{(l)} \neq \pi_\dagger(s_t^{(l)}) \right] \right)$$

$$\leq \frac{k_1 \cdot (\|R\|_\infty + 1 + \lambda)}{L} + \frac{\sum_{l=1}^{k_1} \sum_{t=1}^{T} |\xi_t^{(l)}|}{L \cdot T} + \frac{\|\widehat{\Delta}\|_\infty + \lambda}{L \cdot T} \sum_{l=k_1+1}^{L} \sum_{t=1}^{T} \mathbb{1} \left[ a_t^{(l)} \neq \pi_\dagger(s_t^{(l)}) \right] \tag{117}$$

Define events $C$ and $D$ as the following

$$C := \{k_1 \leq k_0\} \tag{118}$$

$$D := \left\{ \sum_{l=1}^{k_0} \sum_{t=1}^{T} |\xi_t^{(l)}| \leq \sigma \cdot k_0 T \cdot \sqrt{2 \log \frac{2k_0 T}{p}} \right\} \tag{119}$$

That is, $C$ is the event that the attacker uses at most $k_0$ learners in the exploration phase, and $D$ is the event that sum of absolute value of noises in first $k_0$ learners is bounded as in (119). Also let $E$ be the event that for all of $L$ learners, the number of $\varepsilon$-suboptimal steps taken is at most $\text{SUBOPT}(T, \varepsilon, \frac{p}{L})$.

We show that under event $F = B \cap C \cap D \cap E$ the bound on the cost holds, and then show that $\mathbb{P}(F) \geq 1 - 4p$. From event $C$ we get

$$\frac{k_1 \cdot (\|R\|_\infty + 1 + \lambda)}{L} \leq \frac{k_0 \cdot (\|R\|_\infty + 1 + \lambda)}{L} \tag{120}$$

Also $C \cap D$ implies

$$\frac{\sum_{l=1}^{k_1} \sum_{t=1}^{T} |\xi_t^{(l)}|}{L \cdot T} \leq \frac{\sum_{l=1}^{k_0} \sum_{t=1}^{T} |\xi_t^{(l)}|}{L \cdot T} \leq \frac{\sigma \cdot k_0 \cdot \sqrt{2 \log \frac{2k_0 T}{p}}}{L} \tag{121}$$

Finally note that under $B$, since $\widehat{\Delta}$ is a solution of (P2), the target policy is an $\varepsilon$-robust optimal for learners in the attack phase. Thus, all the steps that learners in the attack phase do not follow the target policy are $\varepsilon$-suboptimal. From event $E$ and Lemma E.2 we get

$$\frac{\|\widehat{\Delta}\|_\infty + \lambda}{L \cdot T} \sum_{l=k_1+1}^{L} \sum_{t=1}^{T} \mathbb{1}\left[a_t^{(l)} \neq \pi_\dagger(s_t^{(l)})\right] \leq (\|\widehat{\Delta}\|_\infty + \lambda) \cdot \frac{\text{SUBOPT}(T, \varepsilon, \frac{p}{L})}{T} \tag{122}$$

$$\leq (\|\Delta_M^*\|_\infty + \lambda + m) \cdot \frac{\text{SUBOPT}(T, \varepsilon, \frac{p}{L})}{T} \tag{123}$$

Putting all the bounds together, we get that under event $F$ we have

$$\text{COST}(T, L) \leq \frac{k_0 \cdot (\|R\|_\infty + 1 + \lambda)}{L} + \frac{\sigma \cdot k_0 \cdot \sqrt{2 \log \frac{2k_0 T}{p}}}{L} + (\|\Delta_M^*\|_\infty + \lambda + m) \cdot \frac{\text{SUBOPT}(T, \varepsilon, \frac{p}{L})}{T} \tag{124}$$

which is the bound in the theorem.

Now note that from Corollary D.1 we have $\mathbb{P}(B) \geq 1 - p$. Lemma E.3 and Lemma E.4 show that $\mathbb{P}(C|B) \geq 1 - p$. Thus, we have $\mathbb{P}(B \cap C) = \mathbb{P}(C|B) \cdot \mathbb{P}(B) = (1 - p)^2 \geq 1 - 2p$. Note that from Hoeffding's inequality we have

$$\mathbb{P}\left( |\xi_t^{(l)}| > \sqrt{2\sigma^2 \log \frac{2k_0 T}{p}} \right) \leq \frac{p}{k_0 T} \tag{125}$$

Applying this lemma to all $k_0 T$ steps of first $k_0$ learners, from union bound we get $\mathbb{P}(D) \geq 1 - p$. Finally, from the definition of SUBOPT and union bound, we have $\mathbb{P}(E) \geq 1 - p$. Thus, we have

$$\mathbb{P}(F) = \mathbb{P}(B \cap C \cap D \cap E) \geq 1 - (1 - \mathbb{P}(B \cap C)) - (1 - \mathbb{P}(D)) - (1 - \mathbb{P}(E)) \geq 1 - 4p \tag{126}$$

which concludes the proof. ∎

## G   Technical Details of Attack with Prior Data

In the remark in Section 4.2, we highlighted a stand-alone application of the attack phase procedure, in which the attacker uses some prior set of observations to do the attack without an exploration phase. Here, we

provide the analysis of this attack. Let $N(s, a)$ be the number of times $(s, a)$ is observed in the data and let $N_{\min} = \min_{s,a} N(s, a)$. Define

$$e(s, a) = 2e_Q + \frac{\varepsilon}{\left[\mu_M^{\pi_\dagger \{s;a\}}(s) - e_\mu\right]^+} - \frac{\varepsilon}{\mu_M^{\pi_\dagger \{s;a\}}(s)}$$

where

$$e_Q = \frac{2u + 2\gamma \cdot R_{\text{range}} \cdot w}{(1 - \gamma)^2 \cdot \sqrt{N_{\min}}} \ , \ e_\mu = \frac{2\gamma \cdot w}{(1 - \gamma)^2 \cdot \sqrt{N_{\min}}}.$$

Here, $R_{\text{range}} = \max_{s,a} R(s, a) - \min_{s,a} R(s, a)$.

We show that with probability of at least $1 - 2p$ the cost of this attack is at most

$$\frac{1}{T} \cdot (\|\Delta_M^* + e\|_\infty + \lambda) \cdot \text{SubOpt}(T, \varepsilon, \frac{p}{L}) \tag{127}$$

*Proof.* If $M \in \mathcal{M}$, which happens with probability at least $1 - p$, from Lemma E.2 we have

$$Q_{\text{high}}^{\pi_\dagger}(s, a) \le Q_M^{\pi_\dagger}(s, a) + e_Q \tag{128}$$
$$V_{\text{low}}^{\pi_\dagger}(s) \ge V_M^{\pi_\dagger}(s) - e_Q \tag{129}$$

Also with similar argument as in (82), we have

$$\mu_{\text{low}}^{\pi_\dagger \{s;a\}}(s) \ge \left[\mu_M^{\pi_\dagger \{s;a\}}(s) - e_\mu\right]^+ \tag{130}$$

Thus, we have

$$Q_{\text{high}}^{\pi_\dagger}(s, a) - V_{\text{low}}^{\pi_\dagger}(s) + \frac{\varepsilon}{\mu_{\text{low}}^{\pi_\dagger \{s;a\}}(s)} \le Q_M^{\pi_\dagger}(s, a) - V_M^{\pi_\dagger}(s) + \frac{\varepsilon}{\left[\mu_M^{\pi_\dagger \{s;a\}}(s) - e_\mu\right]^+} + 2e_Q \tag{131}$$

$$= Q_M^{\pi_\dagger}(s, a) - V_M^{\pi_\dagger}(s) + \frac{\varepsilon}{\mu_M^{\pi_\dagger \{s;a\}}(s)} + e(s, a) \tag{132}$$

which gives

$$e(s, a) \ge \left(Q_{\text{high}}^{\pi_\dagger}(s, a) - V_{\text{low}}^{\pi_\dagger}(s) + \frac{\varepsilon}{\mu_{\text{low}}^{\pi_\dagger \{s;a\}}(s)}\right) - \left(Q_M^{\pi_\dagger}(s, a) - V_M^{\pi_\dagger}(s) + \frac{\varepsilon}{\mu_M^{\pi_\dagger \{s;a\}}(s)}\right) \tag{133}$$

$$\ge \left[Q_{\text{high}}^{\pi_\dagger}(s, a) - V_{\text{low}}^{\pi_\dagger}(s) + \frac{\varepsilon}{\mu_{\text{low}}^{\pi_\dagger \{s;a\}}(s)}\right]^+ - \left[Q_M^{\pi_\dagger}(s, a) - V_M^{\pi_\dagger}(s) + \frac{\varepsilon}{\mu_M^{\pi_\dagger \{s;a\}}(s)}\right]^+ \tag{134}$$

$$= \widehat{\Delta}(s, a) - \Delta_M^*(s, a) \tag{135}$$

As $\widehat{\Delta}$ is a solution of (P2), the target policy is $\varepsilon$-robust optimal for the learner. Consequently, the steps in which the target policy is not followed are $\varepsilon$-suboptimal and with probability of at least $1 - p$ at most $\text{SubOpt}(T, \varepsilon, \frac{p}{L})$ for all the learners. Thus, by a union bound, with probability of at least $1 - 2p$ we have

$$\text{Cost}(T, L) = \frac{1}{L \cdot T} \sum_{l=1}^{L} \sum_{t=1}^{T} \left(|r_t^{(l)} - r_t'^{(l)}| + \lambda \mathbb{1}\left[a_t^{(l)} \ne \pi_\dagger(s_t^{(l)})\right]\right) \tag{136}$$

$$\le \frac{1}{L \cdot T} \cdot (\|\widehat{\Delta}\|_\infty + \lambda) \cdot L \cdot \text{SubOpt}(T, \varepsilon, \frac{p}{L}) \tag{137}$$

$$\le \frac{1}{T} \cdot (\|\Delta_M^* + e\|_\infty + \lambda) \cdot \text{SubOpt}(T, \varepsilon, \frac{p}{L}) \tag{138}$$

which proves the claim. ∎

