# OpenReview forum: "Reward Poisoning in Reinforcement Learning: Attacks Against Unknown Learners in Unknown Environments"
_TMLR — Withdrawn by Authors_

### Review · Reviewer_HCvQ · 2023-02-21

**Summary Of Contributions:**

The authors consider the problem of adversarially corrupting the rewards in a reinforcement learning
setting so that a regret minimising policy will play a specific policy rather than learning the
optimal policy. The attack is 'black-box' in the sense that the attacker does not need any knowledge
about the MDP or the learners interacting with it beyond that the latter are suitably well-behaved.
Roughly, it is assumed that the learners have non-trivial sample-complexity.

What is a bit unusual is that there is a sequence of learners interacting with an environment. This is
exploited in the analysis along with a kind of ergodicity assumption because it allows the attacker to
learn the MDP using a relatively naive random corruption and then switch to a white-box attack
that knows (approximately) the MDP.

The basic idea is as follows:

(1) Spend a period of time where the rewards are corrupted to be uniformly sampled from {-1,1} and prove
that under an ergodicity assumption this leads to exploration of all state-action pairs for any well-behaved
algorithm.

(2) Use the learned model and a previously analysed white-box attack for subsequent learners.

The main result is a method of attack for which the cost of attack is bounded in terms of the sample complexity of the learning algorithms and MDP-dependent quantities.

**Audience:**

Yes

**Broader Impact Concerns:**

This is primarily a theoretical work. I have no particular impact concerns and find the authors' disclaimer reasonable.

**Claims And Evidence:**

No

**Requested Changes:**

Unless I misunderstood something, then the setting needs to be modified in some way. It is important that whatever assumptions are made on the learner are satisfied by some existing algorithms or could reasonably be satisfied by future algorithms.

The list of minor comments should also be addressed, but none are critical.

**Strengths And Weaknesses:**

****************************************
Strengths:
****************************************

Robustness to adversarial attack is certainly relevant at the moment, so I broadly support works of this kind.
It's obviously beneficial to develop attacks that do not rely on knowledge of the environment or learner.

****************************************
Weaknesses:
****************************************

In its current form the assumptions on the algorithm and interaction protocol are either (a) not quite accurate; or (b)
not written precisely. These issues make the work unsuitable for publication in its current form, though I am certain
that with some modifications (or maybe clarifications) the results can be made correct.

To begin, it needs to be clarified when the environment resets to the initial state. I cannot see where it is written, but I believe
this happens each time a new learner starts interacting with the MDP. This is important because the assumption on the reachability
of states under any policy are defined relative to the initial state.

Next, Assumption 2.1 needs to be written more precisely. What does it mean to "find an alpha-optimal policy in T
steps with probability of at least 1 - beta"? The parameters in this assumption need to be spelled out. For what
values of epsilon and delta must the sample complexity bound holds. Many algorithms have bounds for any epsilon, but only
fixed delta, but many algorithms also only work for a single pre-specified epsilon, which I believe is not sufficient for
your usage.

The question of "finding" an alpha-optimal policy is especially important. E.g., consider an MDP with three states s_0, s_1, s_2. All
actions in s_0 lead to {s_1,s_2} with equal probability while all actions in {s_1,s_2} lead deterministically to s_2. So after 2 rounds
all algorithms will be in s_2. The reachability conditions on  this MDP are satisfied, but there is no reason why any algorithm
should be able to learn an optimal policy in this MDP because they only visit s_1 at most once.

So there is an inconsistency between the assumption on "finding" an alpha-optimal policy and the claim after the assumption that
many existing algorithms satisfy the claim. What these algorithms (or discounted versions of them) can do is play
actions that are epsilon-optimal along the observed trajectory.

In short, I do not know of any algorithm that provably satisfies Assumption 2.1 under your assumptions and interaction protocol.

You have a few options to fix this:
(1) Switch to an undiscounted episodic setting.
(2) Make reachability assumptions global, rather than relative to an initial state.

Maybe there are others, which I would be happy to hear about.

****************************************
Other questions and comments:
****************************************

* Let's say you make changes (1) or (2) proposed above. Would it not be simpler to explore by setting the reward r(s, a) = 1
for some given (s, a) pair and 0 for all other state/action pairs. Then any regret minimiser will have to find (s,a) to have
small regret/sample-complexity. Once enough (s,a) transitions have been collected switch to a new (s,a) pair. Seems like this idea
may be very easy to analyse and as far as I can see is not theoretically worse. Maybe it works less well
empirically (but maybe better - I don't know).

* There's something a little unsatisfying about the framework because the negative result holds for all regret-minimising algorithms.
So what has been shown is that the goal of regret minimisation and robustness to this level of attack are incompatible. But it seems
to me this attacker has a *lot* of power. Compare to NN attacks where images/covariates are perturbed by a tiny amount to yield
dramatic changes in labels. Here the perturbations are such that the learner is acting in a completely different MDP. I have no suggestions
here, but I am a little skeptical this kind of attack will ever be used.

* I wonder if the reachability assumptions are really needed? E.g., if you use the suggestion in the first bullet above, then
reasonable policies will eventually explore all suitably reachable states while nearly unreachable states should contribute in
only a minor way to the value function.

****************************************
Clarity
****************************************

Generally the paper is quite well written with a good literature summary and largely good intuition. The specific details
of the interaction protocol need to be polished. There is the usual problem in this kind of work that there are many many
constants that are hard to remember. A table summarising them and their Big-O would be most helpful.

****************************************
Technical correctness
****************************************

I did not read the proofs in detail. I have a good feeling that once the setting has been corrected, then some version of
this analysis can be made to work. To be clear. I do not believe that there is a mistake in the present work. Rather that
one of the following holds:

(1) The assumption on the learner is much stronger than the authors intend and such that no existing algorithm satisfies the
requirements; or
(2) There is a mistake in the present analysis.

I rather believe (1) is true and that this can be fixed by changing the setting slightly.

****************************************
Overall:
****************************************

The assumptions and setting need to be clarified. At the moment I do not believe there exist learning algorithms satisfying the given
assumptions. This should be correctable by modifying slightly the setting but would require dramatic change to the manuscript.

****************************************
Minors:
****************************************

* p2. "any efficient RL algorithm" -> "any sample efficient RL algorithm"?

* p2. The definition of an action being called epsilon-suboptimal is a little odd. I would say a_t is epsilon suboptimal if
Q^*(s_t, a_t) <= V^*(s_t) - epsilon. But here the action

* p3. "can "recover" from the change of rewards functions" means what precisely? Maybe some sort of non-stationary regret bound? Refs?

* Remark after Lemma 3.2. Are you sure this is true? There exist norms such that are not monotone
coordinate-wise. E.g., ||(2,1)|| <= ||(1,1)|| can happen for some norms (Mahanoblis with suitable covariance, for example).

* Only stationary deterministic policies are defined. I did not dig carefully into the analysis to make sure the authors are
allowing the learners to use non-stationary (and possibly random policies). I think it is Ok, but the authors should check they
do not need to define other types of policies. E.g., the state/action distribution is defined using stationary deterministic
policies but are surely used for non-stationary random policies. Luckily the deterministic ones are somehow the worst case, but is
it written?

* It would be nice to give as much dependence on fundamental quantities in Big-O notation as possible. E.g., a Big-O expression
that includes the dependence on epsilon, the state/action-space and the minimal visitation probabilities would be welcome.

* Comment after Assumption 2.1. Many of these algorithms are designed for the episodic setting, so I am not sure what it means
for them to satisfy the conditions of the assumption in the discounted setting.

* Let us say X is SG(sigma) if E[exp(L X)] <= exp(L^2 /(2sigma^2)) for all L. Some people call this sigma^2-subgaussian and some
call it sigma-subgaussian. The latter is I believe slightly more usual because it is homogeneous. But maybe it is worth defining
the convention you are using.

---

### Review · Reviewer_Trxu · 2023-03-02

**Summary Of Contributions:**

This paper deals with the problem of reward poisoning, where an adversary modifies the rewards seen by an agent to force it to learn the policy that the adversary wants.

To be more specific, this paper looks at reward poisoning in a black-box (where the adversary does not know the MDP), and federated (there are multiple learners interacting with the same MDP) setting. It does so by modifying a white-box attack procedure by first forcing the agent to explore the entire MDP --- allowing the adversary to estimate a confidence set that will likely contain the true MDP --- and then poisoning the reward using the same approach as used by the white-box poisoning attack such that the agent learns a robust optimal policy for any MDP in this confidence set in the worst case.

The main technical contribution of this work seems to be the exploration induced by this adversary in order for it to then use the white-box attack.

Experiments in a simple grid world show how this black-box algorithm -- U2 -- compares to the white-box approach.

**Audience:**

Yes

**Broader Impact Concerns:**

This work which studies how best to attack RL agents should have some broader impact considerations. The study of these attacks and possible ways they might be helpful or can be mitigated is important. But a short section which allows readers to ruminate on these implications would add value to the paper.

**Claims And Evidence:**

Yes

**Requested Changes:**

* In Sections 3 and 4, equations or symbols which have not been introduced yet are referenced. Examples are the reference of Equation 8 in Section 3.2 which is defined in Section 4.3 and references to $\tilde{P}$ and $\tilde{R}$ in Section 3.2 which are defined in Section 4.1. This makes reading the paper very jarring. The authors should rearrange the text to define required terms before usage, or only give a high level idea in previous sections which is technically clarified in later parts.
* Does the focus on deterministic policies limit the analysis? Perhaps some clarification as to how stochastic policies can be handled would be helpful.
* Should the definition of $\Delta : \mathcal{S} \times \mathcal{A} \longmapsto \mathbb{R}$ be instead limited to positive reals ( $\Delta : \mathcal{S} \times \mathcal{A} \longmapsto \mathbb{R}^{+}$ )?
* Just before Lemma 4.2 `Weissman et al., 2003` is cited in parenthesis, which does not make grammatical sense. Remove them.

**Strengths And Weaknesses:**

## Strengths:
* The addition of the exploration phase to the poisoning attack is straightforward and elegant.
* The problem setup and analysis is neat and methodical.
* The experiment in the grid world domain seems to bear out that the approach works as intended.

## Weaknesses:
* The technical reason for needing multiple learner agents is not clarified to satisfaction. In the experimental results it is clear that having more learners helps the adversary (presumably because it brings down the average cost and allows more exploration), but the justification of federated learning seems like a stretch. A single adversary attacking multiple (perhaps distributed) learners does not seem like a clear motivation, nor does the analysis give a satisfactory technical reason for the inclusion.
* Theorem 3.1 gives the domain of $p$ as $p \in (0, 1)$, but the analysis in this theorem holds for a probability of $ (1 - 4p)$, meaning values of $p > 0.25$ will not be valid. This range should be fixed or clarified.

---

### Review · Reviewer_czcS · 2023-04-10

**Summary Of Contributions:**

The paper studies reward poisoning attacks against reinforcement learning (RL). The authors consider a black-box setting where the attacker does not have any prior knowledge of the reward and transition function of the environment as well as the learning algorithm used by the agent. To develop their attack, called U2, the authors focus on a population learning scenario and make a general and realistic assumption that the RL algorithm used by the agent is sample efficient. The algorithm proceeds in two steps: (i) the exploration phase where the attacker provides rewards sampled from the uniform distribution to compute an estimation of the environment, and (ii) the attack phase: leveraging the learned environment model, the attacker computes the corruption on reward based on ideas from robust policy evaluation. In evaluation, the authors consider a tabular $3\times 3$ grid environment with $4$-actions and Q-learning with UCB exploration algorithm. The results show the effectiveness of U2 in the proposed setting.

**Audience:**

Yes

**Broader Impact Concerns:**

* The authors briefly discuss the potential negative impacts of their attack in Section 6.

**Claims And Evidence:**

No

**Requested Changes:**

* When comparing against Sun et al., the authors claim that "In comparison, our work provides a more general and theoretically sound black-box attack strategy against any efficient RL algorithms, including policy gradient algorithms." I am not sure if U2 works in the deep setting considered by Sun et al., where the assumptions made by the author do not hold (e.g., one cannot do " each iteration of robust policy evaluation involves S special linear programming problems"). I would suggest the authors either revise this claim or provide empirical evidence in the deep setting.

* The authors should motivate the practicality of their attack based on my comments above.

* Demonstrate empirical efficiency of the U2 in more diverse settings as mentioned above and provide an ablation study.

* How does the magnitude of the corrupted rewards differ from the non-corrupted ones, e.g., if the natural reward range is [0-10] in an environment, can your attack provide a reward of 15 or -5 making it easy to be detected?

**Strengths And Weaknesses:**

Strengths:

* Reward poisoning in the RL setting can expose critical vulnerabilities of state-of-the-art learning algorithms wrt imperfect reward modeling or malicious agents and motivate the development of more robust RL algorithms. Despite its importance, there does not exist sufficient work in this direction. The paper advances the state-of-the-art in this direction by considering a black-box setting that reduces assumptions on attacker capabilities.

* The paper is well-organized, and the authors state their assumptions rigorously.  While the technical details can be hard to follow for a non-expert, the high-level technical ideas in Section 3 should be accessible to a broad audience.

* The algorithm developed here is novel and backed up by theoretical evidence.

Weaknesses:

* While the black-box setting considered here is realistic, the attack U2 still relies on the unrealistic cooperation between the agents and the attacker. I cannot think of any real-world setting where the attacker is not only allowed to directly give a random reward to the agents without the agents detecting it but the attacker also has significant computational resources to compute a model of the environment not accessible to the learning agents. Further, in the attack phase, the attacker can run robust policy evaluations to compute a feasible attack. Do the agents stop learning between the exploration and attack phase to allow the attacker to change its reward? The authors do not motivate why this type of cooperation is realistic. As a result, the attack only exposes a theoretical limitation of existing algorithms and not a practical shortcoming of existing RL algorithms.


* The experimental evaluation is quite limited and only considers one relatively toyish scenario. I am not sure if it scales to more complex environments with larger state and action spaces. It will be good to evaluate the effectiveness of U2 in other environments with larger state and action spaces, different learning algorithms, and diverse choices of the target policy. Further, an ablation study on the impacts of different hyperparameters (such as m, $\alpha$, $\lambda$, etc. ) on the attack efficiency will make the evaluation much more comprehensive.

---

### Note · Authors · 2023-04-21

**Comment:**

We would like to thank the reviewers for carefully reviewing our paper! Based on the feedback, we realized that the current assumption that the learner is able to find a near-optimal policy might not be fulfilled when considering the general family of infinite-horizon MDPs. In turn, we would need additional assumptions on the environment for the proposed attack to work or need to consider a different family of  MDPs (e.g., finite-horizon MDPs). We thank the reviewers again for their valuable comments and feedback.


**Withdrawal Confirmation:**

I have read and agree with the venue's withdrawal policy on behalf of myself and my co-authors.